# CROMA: Remote Sensing Representations with Contrastive Radar-Optical Masked Autoencoders

**Anthony Fuller[1,*], Koreen Millard[2], James R. Green[1]**
[1]Department of Systems and Computer Engineering
[2]Department of Geography and Environmental Studies
Carleton University, Ottawa, Canada

## Abstract

A vital and rapidly growing application, remote sensing offers vast yet sparsely labeled, spatially aligned multimodal data; this makes self-supervised learning algorithms invaluable. We present CROMA: a framework that combines contrastive and reconstruction self-supervised objectives to learn rich unimodal and multimodal representations. Our method separately encodes masked-out multispectral optical and synthetic aperture radar samples—aligned in space and time—and performs cross-modal contrastive learning. Another encoder fuses these sensors, producing *joint* multimodal encodings that are used to predict the masked patches via a lightweight decoder. We show that these objectives are complementary when leveraged on spatially aligned multimodal data. We also introduce X- and 2D-ALiBi, which spatially biases our cross- and self-attention matrices. These strategies improve representations and allow our models to effectively extrapolate to images up to $17.6\times$ larger at test-time. CROMA outperforms the current SoTA multispectral model, evaluated on: four classification benchmarks—finetuning (avg.$\uparrow 1.8\%$), linear (avg.$\uparrow 2.4\%$) and nonlinear (avg.$\uparrow 1.4\%$) probing, $k$NN classification (avg.$\uparrow 3.5\%$), and $K$-means clustering (avg.$\uparrow 8.4\%$); and three segmentation benchmarks (avg.$\uparrow 6.4\%$). CROMA's rich, optionally multimodal representations can be widely leveraged across remote sensing applications.

## 1 Introduction

Deep learning has led to rapid advances in remote sensing, augmenting our ability to understand and monitor our planet. The remote sensing community has developed many application-specific deep learning models, specifically for satellite imagery: identifying heavily polluting brick kilns [1, 2] or illegal airstrips [3]; monitoring deforestation [4, 5, 6, 7] or crops [8, 9, 10]; detecting floods [11, 12] or wildfires [13, 14, 15]; even estimating household income [16, 17] or poverty [18, 19, 20]. Deep learning-based remote sensing is playing a growing role in tackling our climate crisis [21, 22, 23]. Recently, researchers leveraged self-supervised learning to pretrain remote sensing models that can be employed on these tasks, and more [24, 25, 26, 27]. Self-supervised methods are invaluable for remote sensing, as there are petabytes of publicly available raw data from which to learn general representations, while only limited annotated data exists for downstream applications.

Self-supervised representations are often learned via contrastive approaches [28, 29, 30] or reconstruction approaches [31, 32, 33]. Contrastive approaches encourage the representations of positive pairs of samples—built by producing another view of a sample, for instance, from another sensor [34] or time [35], or by augmentations [30]—to be similar, and the representations of negative pairs to be dissimilar; this process can learn descriptive, object-focused representations. Models trained with a

---

*All correspondence should be addressed to Anthony Fuller: `anthony.fuller@carleton.ca`

contrastive objective learn to discard information not shared between views [36, 37], regardless of the information's usefulness on downstream tasks; this makes the representations they learn sensitive to how positive pairs are built [30, 38, 39, 40]. Conversely, models trained with reconstruction or autoencoding objectives—for instance, predicting hidden pixels [31, 41]—learn to capture as much information as possible [42, 43, 44]. Reconstruction approaches do not rely on multiple views and scale incredibly well [31, 45, 46], but they learn representations that require significant finetuning to be useful on downstream tasks [43, 45]. Park et al. [47] show vision transformers (ViTs [48]) trained with contrastive learning focus more on shapes and low-frequency information than ViTs trained with reconstruction approaches, which focus more on textures and high-frequency information. They show that combining both objectives may achieve a sweet spot that learns better representations than either objective alone. Several other frameworks have been developed that leverage both objectives to learn SoTA representations [49, 50, 51, 52]; however, none are designed for spatially aligned multimodal data.

Researchers developing foundation models for remote sensing have yet to take advantage of the multimodal data ubiquitous to remote sensing. For instance, the Sentinel missions—imaging the Earth's landmass multiple times per month since 2015—consist of multispectral optical imagery acquired by Sentinel-2 and Synthetic Aperture Radar (SAR) data acquired by Sentinel-1. By exploiting differences in how electromagnetic radiation interacts with Earth surface materials and measuring the radiation at many wavelengths (ranging from 440 to 2,200 nm), Sentinel-2 multispectral optical imagery can be used to characterize the material composition of objects [53]. By actively transmitting and receiving longer wavelength (5.5 cm) electromagnetic pulses, Sentinel-1 SAR can be used to characterize the geometry, roughness, and electrical properties of objects [54]. These modalities have proven complementary across remote sensing applications [55, 56, 57]. Importantly for our work, they are spatially aligned, allowing multiple views of the same feature on the ground. Moreover, because data from the Sentinel missions are freely available, they have become the most widely used source of satellite imagery in research; thus, models with useful representations of Sentinel-1 & 2 imagery can be immediately leveraged in scientific research, improving our ability to understand our planet.

These observations motivate $\underline{\text{C}}$ontrastive $\underline{\text{R}}$adar-$\underline{\text{O}}$ptical $\underline{\text{M}}$asked $\underline{\text{A}}$utoencoders (**CROMA**): a framework for learning rich representations of multimodal, spatially aligned, 2D data; which we leverage to pretrain ViT models on Sentinel-1 & 2 data, providing the most valuable foundation models for Earth Observation, to date. We highlight three contributions: ❶ CROMA significantly outperforms the current SoTA multispectral model, SatMAE [26], under an *extensive* evaluation. ❷ CROMA learns representations that are *optionally* multimodal (i.e., they can be effectively leveraged when either or both modalities are available) and *extrapolate* to larger images at test-time (i.e., they can be effectively leveraged on images larger than those on which the model was trained). ❸ CROMA's effectiveness is driven by two innovations: our complementary pretraining objectives and our novel relative position encoding (RPE) strategies that contribute to the quality of learned representations.

## 2 Method

In this section, "optical" refers to 12-channel multispectral optical imagery acquired by Sentinel-2, and "radar" refers to 2-channel SAR backscatter data acquired by Sentinel-1.

**Architecture Background.** Most work combining contrastive and reconstruction objectives learning *joint* multimodal representations is in the image-text domain [51, 58, 59, 60, 61, 62, 63, 64, 65]; heavily inspiring our architecture, Contrastive Captioning (CoCa [49]) learns SoTA image-text representations that are optionally multimodal. The CoCa framework consists of two unimodal encoders—one for images, one for text—and a multimodal decoder that receives text encodings at the bottom of the network and cross-attends to image encodings. CoCa is trained with two objectives: an image↔text contrastive objective between unimodal encoders and an image-captioning objective at the output of the multimodal decoder. Our framework significantly adapts CoCa to aligned multimodal 2D data by masking both modalities, introducing a multimodal encoder, a lightweight decoder (only used during pretraining, inspired by masked autoencoders [31]), and novel cross-attention and self-attention positional biases.

**Model Architecture.** CROMA consists of three encoders (Fig. 1). ❶ A unimodal radar ViT $f_R$ that encodes radar inputs $I_R \in \mathbb{R}^{2 \times H \times W}$ into $L$ patch encodings $\mathcal{E}_R \in \mathbb{R}^{L \times D}$, i.e., $\mathcal{E}_R = f_R(I_R)$. ❷ A unimodal optical ViT $f_O$ that encodes optical inputs $I_O \in \mathbb{R}^{12 \times H \times W}$ into $L$ patch encodings

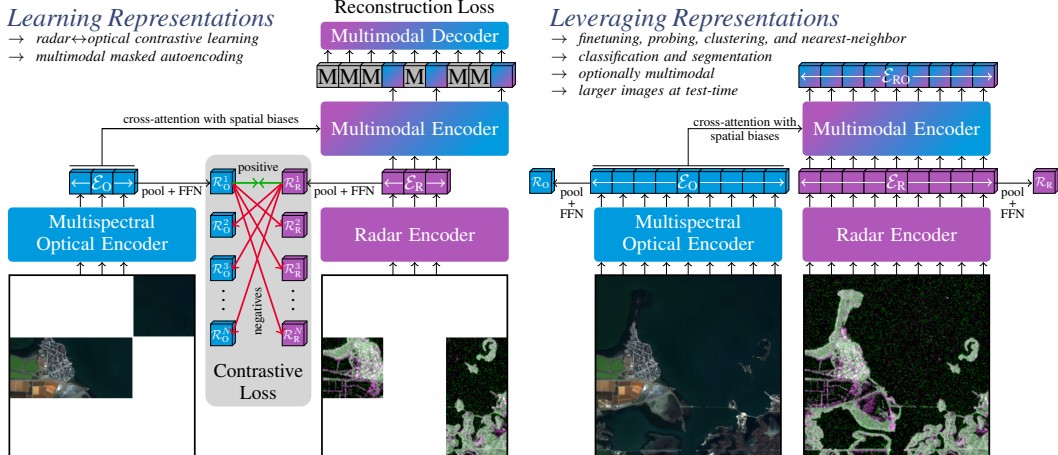

**Figure 1: (Left)** Our CROMA framework jointly leverages radar↔optical contrastive learning and masked autoencoding to learn rich, self-supervised representations. **(Right)** Leverage representations on: unimodal or multimodal data, larger images at test-time, and diverse tasks and methods.

$\mathcal{E}_O \in \mathbb{R}^{L \times D}$, i.e., $\mathcal{E}_O = f_O(I_O)$. ❸ A multimodal radar-optical transformer $f_{RO}$ that encodes $L$ radar-optical patches, $\mathcal{E}_{RO} \in \mathbb{R}^{L \times D}$, i.e., $\mathcal{E}_{RO} = f_{RO}(\mathcal{E}_R, \mathcal{E}_O)$. For each set of unimodal encodings, we build full-image representations $\mathcal{R} \in \mathbb{R}^D$ by processing the mean pooled patch encodings through a feedforward network (FFN), i.e., $\mathcal{R}_R = \text{FFN}_R(\text{MeanPool}(\mathcal{E}_R))$ and $\mathcal{R}_O = \text{FFN}_O(\text{MeanPool}(\mathcal{E}_O))$. Our patch size is $8 \times 8$ pixels for both modalities, and our default image size is $120 \times 120$ pixels. Our radar encoder has $N/2$ transformer layers, and our optical encoder has $N$ transformer layers ($N$ is 12 for ViT-B and 24 for ViT-L backbones). All unimodal encoder layers are composed of self-attention and FFN sublayers. Our multimodal $N/2$-layer encoder—composed of self-attention, cross-attention, and FFN sublayers—encodes both modalities into a *single* sequence of $L$ patch encodings; this encoder receives radar patch encodings at the bottom of the network and learns multimodal representations by cross-attending to optical patch encodings. Multimodal representations can be built via pooling multimodal patch encodings, i.e., $\mathcal{R}_{RO} = \text{MeanPool}(\mathcal{E}_{RO})$. Our ViT backbones do not use sinusoidal position embeddings; instead, we bias the self-attention and cross-attention matrices with the distances between patches.

**ALiBi Background.** ALiBi [66] is a simple and intuitive RPE method for transformers that biases the self-attention matrix based on the distance between tokens in a 1D sequence. Each self-attention head receives positional biases with different strengths, called slopes $m$. With 16 attention heads, the geometric sequence defines these scalar slopes starting at $\frac{1}{\sqrt{2}}$, i.e., $\frac{1}{2^{0.5}}, \frac{1}{2^1}, \frac{1}{2^{1.5}}, \dots, \frac{1}{2^8}$. Biases are subtracted from the attention matrix before the softmax is calculated. Specifically, the pre-softmax attention matrix $A \in \mathbb{R}^{h \times L \times L}$ ($h$ is the number of heads, $L$ is the sequence length) is populated with attention scores $a_{hij}$ for the $i$th query $q_{hi} \in \mathbb{R}^d$ and $j$th key $k_{hj} \in \mathbb{R}^d$ ($d$ is the head dimension): $a_{hij} = \sqrt{d} \cdot q_{hi} \cdot k_{hj}$, without ALiBi; and $a_{hij} = \sqrt{d} \cdot q_{hi} \cdot k_{hj} - \text{distance}(i,j) \cdot m(h)$, with ALiBi. Crucially, ALiBi does not add position embeddings at the bottom of the network; instead, the relative positions between tokens are encoded in the attention matrix itself. To date, ALiBi is the only position encoding method for transformers that has been demonstrated to extrapolate at test-time to sequences far longer than those on which it was trained.

**2D-ALiBi and X-ALiBi.** We extend ALiBi to 2D inputs by biasing the self-attention matrix based on the Euclidean distance between query-key pairs in a ViT—we call this 2D-ALiBi (Fig. 2). We also extend ALiBi to cross-attention by biasing the cross-attention matrix based on the Euclidean distance between *cross-modal* query-key pairs—we call this X-ALiBi. (In cross-attention [67], queries are built from the previous layer, whereas keys and values are built from optical encodings.) For both 2D and X-ALiBi, we calculate our attention-head slopes with the same geometric sequence as ALiBi. Since our two modalities

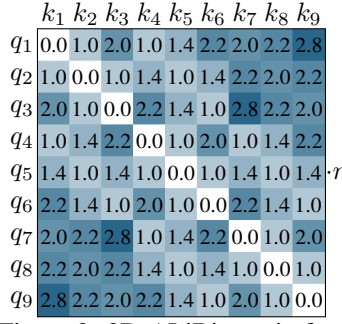

Figure 2: 2D-ALiBi matrix for an image with 9 patches ($3^2$ before flattening); $q$ is for query, $k$ is for key, and $m$ is a fixed scalar that biases attention heads at different rates.

are aligned 2D sensor data, our 2D-ALiBi and X-ALiBi matrices are identical. The primary motivation of 2D-ALiBi is to learn representations that can generalize across image sizes; this is particularly useful in remote sensing and is likely useful in other domains. The primary motivation of X-ALiBi is to improve sensor fusion by inserting positional information in the cross-attention sublayer of our multimodal encoder, $f_{\text{RO}}$. These position encoding techniques are rotation and translation invariant, which are desirable properties for the overhead imagery in Earth Observation.

**MAE Background.** A masked autoencoder (MAE [31]) rearranges an image into a sequence of non-overlapping patches, then randomly samples a large portion of patches to be held-out. The "visible" patches, that are *not* held-out, are encoded by a ViT. This MAE-style masking is ingenious: it leverages sparse computation while only requiring dense operations that run efficiently on modern hardware. MAE introduces a lightweight decoder that receives visible patch encodings and hidden mask embeddings, which are both added to 2D-sinusoidal position embeddings. The decoder outputs predictions of the pixel values of the held-out patches. Both the encoder and decoder are pretrained end-to-end to minimize the mean squared error between patch predictions and the originally held-out patches. The pretrained encoder can then be leveraged on downstream applications.

**Reconstruction Objective.** We independently mask 75% of radar and optical patches and encode the unmasked patches with our three encoders, i.e., $\mathcal{E}_{\text{R}}^{um} = f_{\text{R}}(\text{I}_{\text{R}}^{um})$, $\mathcal{E}_{\text{O}}^{um} = f_{\text{O}}(\text{I}_{\text{O}}^{um})$, and $\mathcal{E}_{\text{RO}}^{um} = f_{\text{RO}}(\mathcal{E}_{\text{R}}^{um}, \mathcal{E}_{\text{O}}^{um})$; where $um$ means unmasked. We introduce a lightweight 1-layer transformer decoder $f_{\text{DEC}}$ that receives multimodal patch encodings and mask embeddings after adding 2D-sinusoidal position embeddings and predicts a target image, i.e., $\hat{\text{I}} = f_{\text{DEC}}(\text{CONCAT}[\mathcal{E}_{\text{RO}}^{um}, \text{Emb}^{\text{mask}}] + \text{Emb}^{\text{positions}})$. We split the channels of $\hat{\text{I}}$ to form predictions for each sensor, $\hat{\text{I}}_{\text{O}}$ and $\hat{\text{I}}_{\text{R}}$; the loss is only applied at the locations of the masked-out patches:

$$\mathcal{L}_{\text{MAE}} = \frac{1}{N} \sum_i^N \left( \underbrace{\frac{\sum_j^M \left( \hat{\text{I}}_{\text{O}}^{ij} - \text{Norm}(\text{I}^{ij})_{\text{O}} \right)^2}{M}}_{\text{optical reconstruction}} + \underbrace{\frac{\sum_j^M \left( \hat{\text{I}}_{\text{R}}^{ij} - \text{Norm}(\text{I}^{ij})_{\text{R}} \right)^2}{M}}_{\text{radar reconstruction}} \right)$$

where $N$ is the batch size, $M$ is the number of masked patches, and $\text{Norm}$ sets the mean to $0$ and standard deviation to $1$ for target patches (following MAE). Along with learning unimodal representations, this objective spatially fuses our two sensors, i.e., it builds multimodal patch encodings that represent information from *both* sensors in the patch, corresponding to an $80\,\text{m} \times 80\,\text{m}$ square on the ground ($8 \times 8$ patches at $10\,\text{m}$ resolution). Finally, 2D and X-ALiBi can be easily adapted to MAE-style masking by removing the masked-out columns and rows from the bias matrix.

**Contrastive Learning Background.** Contrastive learning aims to classify the correct pairings of samples derived from a batch. Logits are formed by measuring the similarity between the projected representations of samples. As a result, the representations of positive pairs are pulled together, and the representations of negative pairs are pushed apart. Very recently, FLIP [68] performs contrastive learning with masked-out representations via MAE-style masking. This speeds up pretraining and enables larger batches due to the reduced memory per sample. FLIP performs on par with CLIP [69]—the foundational work that learns rich representations via image↔text contrastive learning—but can be pretrained at half the cost.

**Contrastive Objective.** We perform radar↔optical contrastive learning across the unimodal representations of our masked-out sensor data using the InfoNCE loss [28]. For an optical anchor image, the positive sample is the geographically and temporally matched radar sample, and the negative samples are all *other* radar samples from the batch; likewise, our radar representations are pulled towards (positives) or pushed apart (negatives) from optical representations:

$$\mathcal{L}_{\text{Con}} = -\frac{1}{2N} \left( \underbrace{\sum_i^N \log \frac{\exp\left( z_{\text{R}}^{i\top} z_{\text{O}}^i / \sigma \right)}{\sum_j^N \exp\left( z_{\text{R}}^{i\top} z_{\text{O}}^j / \sigma \right)}}_{\text{radar-to-optical}} + \underbrace{\sum_i^N \log \frac{\exp\left( z_{\text{O}}^{i\top} z_{\text{R}}^i / \sigma \right)}{\sum_j^N \exp\left( z_{\text{O}}^{i\top} z_{\text{R}}^j / \sigma \right)}}_{\text{optical-to-radar}} \right)$$

where $z_{\text{R}}$ and $z_{\text{O}}$ are $\ell_2$ normalized linear projections of radar and optical representations, respectively, i.e., $z_{\text{R}} = \text{Norm}(\text{Linear}_{\text{R}}(\mathcal{R}_{\text{R}}))$ and $z_{\text{O}} = \text{Norm}(\text{Linear}_{\text{O}}(\mathcal{R}_{\text{O}}))$. $\sigma$ is the softmax temperature, and

$N$ is the batch size. Crucially, we only encode a small portion of input patches, which form our representations. This masking provides advantages: it enables larger batches, speeds up pretraining, and enables our multimodal reconstruction objective with the same computational graph. This radar↔optical contrastive objective encourages representations to be sensor-invariant, i.e., to capture information shared *between* sensors.

**Combined Pretraining Objective.** We combine contrastive learning and masked sensor modeling pretraining objectives: $\mathcal{L} = \lambda_{\mathrm{Con}}\mathcal{L}_{\mathrm{Con}} + \lambda_{\mathrm{MAE}}\mathcal{L}_{\mathrm{MAE}}$. We set both task weights (i.e., $\lambda_{\mathrm{Con}}$ and $\lambda_{\mathrm{MAE}}$) to 1 and ablate them in Appendix §A.1.

## 3    Experiments

**Pretraining.** We pretrain CROMA models on the SSL4EO dataset [70]—a large geographically and seasonally diverse unlabeled dataset. SSL4EO consists of 1 million paired Sentinel-1 GRD & Sentinel-2 L2A samples of $264 \times 264$ pixels. Sentinel-1 channels consist of VV and VH backscatter. Sentinel-2 channels consist of 12 surface reflectance multispectral bands (the cirrus band is removed). We pretrain CROMA-B (ViT-B backbone) for 300 epochs and CROMA-L (ViT-L backbone) for 600 epochs. Crucially, a single pretraining run trains all three encoders (optical, radar, and joint radar-optical) end-to-end; users can then finetune one or multiple encoders, depending on their task and data availability. We perform all pretraining experiments on an NVIDIA DGX server ($8\times$ A100–80 GB), including ablations. Please see Appendix §A.3 for more details.

**Comparisons.** We compare CROMA to all available multispectral optical foundation models, which include two models pretrained by [71] using radar↔optical contrastive learning; two models pretrained by [70] using the MAE [31] and DINO [72] frameworks; and two models pretrained by [26] using their multispectral representation learning framework, SatMAE. We also compare CROMA to a SoTA method for learning visual representations of natural images—image joint embedding predictive architecture (I-JEPA, [73])—that we leverage to pretrain a ViT-B model on SSL4EO's optical imagery for 300 epochs. To enable a fair comparison between models, we evaluate all models under identical conditions and hyper-parameter budgets (please see Appendix §A.4.2 for details). This is necessary because the originally reported results of these models occurred under inconsistent evaluation conditions—for instance, data splits or training data amounts. We use the latest publicly available models for all evaluations and preprocess data according to official repositories. For radar and radar-optical datasets, we compare CROMA to SatViT-V2 [74], a model pretrained using MAE [31] on stacked Sentinel-1 & 2 imagery; and DeCUR, a model pretrained—concurrently with this work—by [75] using their multimodal representation learning framework.

### 3.1    Multispectral Optical Experiments

**Classification Setup.** We evaluate CROMA by finetuning, frozen linear and nonlinear probing, $k$NN classifying, and $K$-means clustering pretrained representations across four Sentinel-2 classification benchmarks. ❶ The multi-label BigEarthNet dataset [76] (35,420 train samples and 118,065 validation samples); this is $10\%$ of the complete BigEarthNet training set that is now used by default [25, 26] to reduce the costs of finetuning and is better suited for a remote sensing benchmark [22]. ❷ The fMoW-Sentinel dataset [26] (71,287 train samples and 84,939 validation samples); this is also $10\%$ of the complete training set. Following BigEarthNet's use, we believe this smaller training set is a more appropriate benchmark for model evaluation, but we show results on the complete training set in Appendix §A.4.1. ❸ The EuroSAT dataset [77] (16,200 train samples and 5,400 validation samples). ❹ The Canadian Cropland dataset [78] (53,884 train samples and 23,088 validation samples); this is a new benchmark inspired by EuroSAT but is more challenging, as the crop types (barley, canola, corn, etc.) can be visually similar. For finetuning and linear probing, we add a linear layer for these tasks atop the full-image representations, i.e., $\mathrm{Linear}(\mathcal{R}_{\mathrm{O}})$. For nonlinear probing, we use an MLP with one hidden 2048-d layer, i.e., $\mathrm{Linear}(\mathrm{ReLU}(\mathrm{Linear}(\mathcal{R}_{\mathrm{O}})))$. Additionally, we perform non-parametric $k$NN classification ($k = 20$) and $K$-means clustering for single-label benchmarks to evaluate frozen representations. [79] shows that no single method of evaluating representations is the best; they recommend including $k$NN and $K$-means alongside linear probing. Other studies [80, 72, 81] show a rank mismatch between $k$NN and linear probing evaluations—indicating they offer complementary estimates of representation quality. Please see Appendix §A.4.1 and A.4.2 for implementation, data splits, and hyper-parameter details.

Table 1: **Classification results on four benchmarks, under finetuning (FT), and frozen linear (LP) and nonlinear (MLP) probing.** * denotes originally reported results; we obtain all other results under identical conditions.

| Method | Backbone | BigEarthNet (10%) mAP FT | MLP | LP | fMoW-Sentinel (10%) Top 1 Acc. FT | MLP | LP | EuroSAT Top 1 Acc. FT | MLP | LP | Canadian Cropland Top 1 Acc. FT | MLP | LP |
|---|---|---|---|---|---|---|---|---|---|---|---|---|---|
| radar↔optical [71] | ResNet50 | 77.65 | 78.79 | 77.44 | 32.03 | 6.46 | 6.01 | 96.31 | 86.35 | 78.81 | 57.44 | 58.09 | 55.55 |
| radar↔optical [71] | Swin-T | 86.41 | 78.65 | 77.93 | 52.01 | 28.54 | 31.06 | 98.09 | 93.50 | 94.78 | 70.98 | 60.36 | 57.05 |
| MAE [31, 70] | ViT-S | 86.15 | 81.70 | 75.94 | 51.79 | 31.70 | 27.69 | 98.78 | 94.46 | 91.80 | 74.02 | 59.07 | 48.38 |
| DINO [72, 70] | ViT-S | 87.04 | 84.96 | 81.58 | 52.79 | 35.62 | 32.64 | 98.63 | 97.07 | 96.07 | 75.27 | 67.35 | 59.94 |
| I-JEPA [73] | ViT-B | 85.92 | 84.27 | 80.80 | 53.54 | 35.76 | 32.35 | 99.20 | 96.60 | 95.63 | 75.13 | 66.69 | 60.17 |
| SatMAE [26] | ViT-B | 85.94 | 83.48 | 79.36 | 57.20 | 37.28 | 35.17 | 99.20 | 97.28 | 96.61 | 73.58 | 66.02 | 60.40 |
| CROMA | ViT-B | 87.58 | 86.29 | **85.04** | 54.47 | 39.67 | 38.42 | 99.22 | 97.89 | 97.59 | 76.17 | 67.62 | 63.39 |
| SatMAE [26] | ViT-L | 86.18/ 82.62* | 84.01 | 80.29 | 58.19 | 38.18 | 36.76 | 99.35/ 98.98* | 97.67 | 97.65 | 74.06 | 67.03 | 61.75 |
| CROMA | ViT-L | **88.29** | **86.46** | 85.01 | **59.02** | **40.07** | **39.17** | **99.46** | **98.04** | **98.02** | **78.07** | **67.94** | **64.02** |

**Classification Results.** CROMA ranks first, averaged across four Sentinel-2 (multispectral optical) benchmarks, under finetuning, frozen linear and nonlinear probing, $k$NN classification, and $K$-means clustering (Table 1, 2). The *only* case where a SatMAE model outperforms a CROMA model of the same backbone is finetuning on the fMoW-Sentinel dataset. However, SatMAE was pretrained on fMoW-Sentinel—meaning there is no distribution shift between pretraining and finetuning. This gives SatMAE an advantage on the fMoW-Sentinel benchmark since downstream performance is impacted by the similarity be-

Table 2: **Non-parametric $k$NN classification and $K$-means clustering results.** * denotes 10% of the training set.

| Method | Backbone | fMoW-Sent.* Top 1 Acc. $k$NN | $K$-means | EuroSAT Top 1 Acc. $k$NN | $K$-means | Can. Crop. Top 1 Acc. $k$NN | $K$-means |
|---|---|---|---|---|---|---|---|
| radar↔optical [71] | ResNet50 | 7.07 | 3.94 | 52.09 | 23.52 | 49.82 | **23.15** |
| radar↔optical [71] | Swin-T | 19.46 | 7.93 | 85.07 | 56.91 | 48.22 | 21.57 |
| MAE [31, 70] | ViT-S | 22.25 | 7.54 | 87.33 | 41.50 | 56.01 | 18.31 |
| DINO [72, 70] | ViT-S | 28.89 | 8.23 | 94.20 | 45.83 | 60.70 | 20.22 |
| I-JEPA [73] | ViT-B | 25.45 | 8.01 | 89.02 | 42.89 | 57.48 | 18.06 |
| SatMAE [26] | ViT-B | 26.76 | 8.98 | 89.28 | 44.87 | 56.20 | 20.14 |
| CROMA | ViT-B | 32.03 | 11.35 | 95.26 | 76.06 | 61.25 | 22.55 |
| SatMAE [26] | ViT-L | 26.77 | 9.17 | 89.57 | 38.48 | 56.93 | 19.65 |
| CROMA | ViT-L | 29.54 | 10.12 | 94.70 | 61.24 | 59.41 | 21.27 |

tween upstream and downstream data [82, 83, 84]. Despite this observation, CROMA-B outperforms SatMAE-B on fMoW-Sentinel under linear probing (↑ 3.3%), nonlinear probing (↑ 2.4%), $k$NN (↑ 5.3%), and $K$-means (↑ 2.4%). Additionally, CROMA models are more than $4\times$ faster during finetuning and inference than their SatMAE counterparts (please see Appendix §A.4.2 for a comparison). On all evaluations, CROMA outperforms a ViT-B model pretrained using the I-JEPA framework on the SSL4EO dataset—I-JEPA is the current SoTA framework for learning self-supervised representations of ImageNet [73]. We also plot UMAP [85] embeddings (Fig. 3); CROMA shows a strong separation of EuroSAT classes.

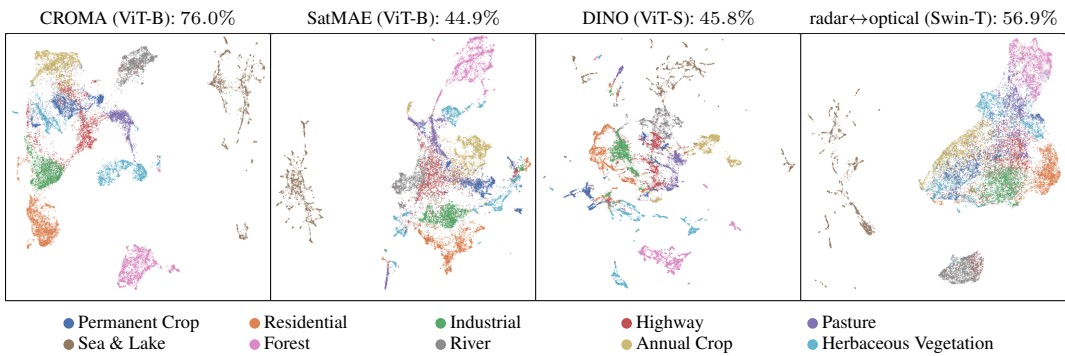

Figure 3: UMAP embeddings and $K$-means clustering accuracies of CROMA (ViT-B), SatMAE (ViT-B) [26], DINO (ViT-S) [70], and radar↔optical (Swin-T) [71] models on EuroSAT [77].

**Sparse Probing.** Inspired by the sparse probing of language models [86], we sparsely probe the 768-dimensional (ViT-B backbone) image representations by restricting the linear probe to $k$ dimensions. For each class, we rank all dimensions and then train a linear probe on the top $k$ to perform binary classification (this setup follows [86], please see Appendix §A.4.2 for more details). This experiment helps us understand the information contained in the learned representations. For example, in Fig. 4 (B), there is a large drop in F1 score when decreasing $k$—indicating that BigEarthNet's "beaches, dunes, sands" class is not well represented by individual dimensions. Rather, this class is represented by a composition of many features that are correlated with beaches. Conversely, BigEarthNet's "agriculture with natural vegetation" class maps well to a *single* dimension of CROMA's representations, achieving an F1 score of $49\%$ when $k = 1$ (Fig. 4 (A)). In Appendix §A.6, we show sparse probing results for all classification tasks and for all classes.

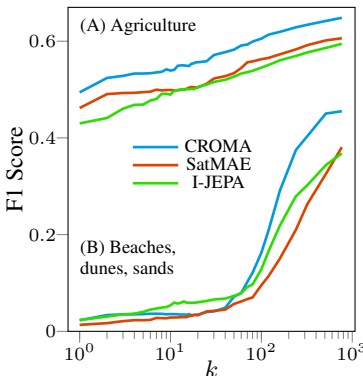

Figure 4: Sparse probing two BigEarthNet classes.

**Segmentation Setup.** We evaluate all ViT-based models on three Sentinel-2 segmentation benchmarks. ❶ The DFC2020 dataset [87] (46,152 train samples and 8,874 validation samples). ❷ A subset of the Dynamic World dataset [88] that was annotated by experts; hence we call it DW-Expert (20,422 train samples and 51,022 validation samples). DW-Expert is a new, high-quality benchmark that was annotated with the help of high-resolution satellite and street-level imagery. ❸ The MARIDA dataset [89] (1,682 train samples and 1,615 validation samples), which is a small, sparsely labeled dataset of marine debris. For all tasks, we linear probe frozen patch encodings, i.e., $\text{Linear}(\mathcal{E}_\text{O})$. We crop images of $96 \times 96$ from the original images ($256 \times 256$ for DFC2020, $510 \times 510$ for DW-Expert, and $256 \times 256$ for MARIDA). We train and evaluate all models on these $96 \times 96$ images, which is the default image size for SatMAE, enabling a fair comparison with the SoTA.

**Segmentation Results.** CROMA outperforms SatMAE by averages of $5.4\%$ and $6.4\%$ for ViT-B and ViT-L backbones, respectively (Table 3). These results demonstrate that CROMA effectively learns fine-grained patch-level features useful for dense prediction tasks like semantic segmentation.

Table 3: Semantic segmentation (mIoU) results on three Sentinel-2 benchmarks.

| Method | Backbone | DFC2020 | DW-Expert | MARIDA |
|---|---|---|---|---|
| MAE | ViT-S | 35.63 | 46.63 | 48.06 |
| DINO | ViT-S | 32.34 | 48.34 | 49.38 |
| I-JEPA | ViT-B | 36.72 | 50.82 | 53.85 |
| SatMAE | ViT-B | 45.53 | 51.03 | 58.17 |
| CROMA | ViT-B | 46.67 | 58.55 | **65.56** |
| SatMAE | ViT-L | 44.13 | 51.50 | 57.12 |
| CROMA | ViT-L | **49.78** | **58.71** | 63.32 |

### 3.2 Radar and Radar-Optical Experiments

**Multimodal Setup.** BigEarthNet and DFC2020 also contain spatially and temporally aligned Sentinel-1 samples alongside Sentinel-2. This allows us to evaluate our multimodal representations and directly compare them with optical-only representations on the same benchmark. We linear probe frozen image representations for BigEarthNet, i.e., $\text{Linear}(\text{CONCAT}(\mathcal{R}_\text{R}, \mathcal{R}_\text{O}, \mathcal{R}_\text{RO}))$, and patch encodings for DFC2020, i.e., $\text{Linear}(\text{CONCAT}(\mathcal{E}_\text{R}, \mathcal{E}_\text{O}, \mathcal{E}_\text{RO}))$. Concurrent with this work, DeCUR [75] learns radar and radar-optical representations via their novel multimodal framework. They fit a linear probe on top of DeCUR's representations on 1% of the BigEarthNet training set and report results on the BigEarthNet validation set; we follow this experimental procedure with our CROMA-B model for a direct comparison.

Table 4: Multimodal (Sentinel-1 & 2) linear probing results on classification and segmentation benchmarks.

| Model | BigEarthNet mAP | DFC2020 mIoU |
|---|---|---|
| SatViT-V2 [74] | 79.80 | 46.20 |
| CROMA-B | **86.24** | 51.58 |
| CROMA-L | 86.20 | **53.24** |

**Multimodal Results.** CROMA significantly outperforms the joint multimodal representations learned by SatViT-V2 [74] (Table 4). Our joint multimodal representations outperform our optical-only representations by $1.2\%$ on BigEarthNet (both CROMA-B and CROMA-L) and by $4.9\%$ (CROMA-B) and $3.5\%$ (CROMA-L) on DFC2020 (Table 1, 3, 4)—justifying our multimodal approach. CROMA's radar-only and radar-optical representations also outperform the concurrent DeCUR under a linear probing experiment (Table 5).

Table 5: Linear probing radar and radar-optical representations.

| | BigEarthNet (1%) mAP | |
|---|---|---|
| Method | Radar | Radar-Opt. |
| DeCUR | 73.7 | 79.4 |
| CROMA | **75.7** | **81.8** |

Table 6: Linear probing ablation results on radar-only ("R"), optical-only ("O"), and joint radar-optical ("RO") inputs. We consider both performance and cost to select our design. All rows below "all default" report the performance differences between ablated cases and the default.

| | Case (default) | Ablation | batch size | cost | Classification (mAP) | | | Segmentation (mIoU) | | | Avg |
|---|---|---|---|---|---|---|---|---|---|---|---|
| | | | | | R | O | RO | R | O | RO | |
| | all default | | 7.2k | 1.0× | 78.2 | 84.5 | 84.8 | 40.8 | 56.0 | 56.5 | 66.8 |
| ❶ | objectives (both) | MAE-only | 7.2k | 1.0× | −10.4 | −6.0 | −5.6 | −9.2 | −8.4 | −5.1 | −7.5 |
| | | contrast-only | 14k | 0.6× | −3.2 | −3.0 | — | −2.9 | −3.7 | — | — |
| ❷ | position encoding (2D-ALiBi + X-ALiBi) | 2D-ALiBi | 7.2k | 1.0× | −0.2 | −0.2 | −0.3 | 0.1 | 0.0 | −0.9 | −0.2 |
| | | PEG [90] | 6.9k | 1.0× | −0.3 | 0.0 | −0.1 | −0.6 | −1.0 | −1.9 | −0.7 |
| | | 2D-sinusoidal | 7.2k | 1.0× | −3.8 | −2.9 | −1.3 | −2.2 | −0.1 | −0.2 | −1.7 |
| ❸ | masking (independent 75%) | shared 25% | 2.4k | 3.0× | −0.5 | −1.0 | −1.1 | 1.1 | 0.5 | −0.2 | −0.2 |
| | | shared 50% | 3.7k | 1.8× | −0.3 | −0.7 | −0.7 | 1.0 | 0.6 | 0.1 | 0.0 |
| | | shared 75% | 7.2k | 1.0× | −0.8 | −0.5 | −0.3 | 0.0 | 0.1 | 0.0 | −0.2 |
| | | independent 50% | 3.7k | 1.8× | −0.2 | −0.5 | −0.5 | 0.9 | 0.7 | 0.3 | 0.1 |
| ❹ | MAE target (radar & optical) | radar-only | 7.2k | 1.0× | −0.1 | −0.2 | −0.6 | 0.3 | −0.2 | −1.4 | −0.5 |
| | | optical-only | 7.2k | 1.0× | −0.1 | −0.2 | −0.2 | −0.1 | −0.1 | −0.6 | −0.2 |
| ❺ | MAE decoder (depth=1, dim=512) | depth=6, dim=768 | 4.3k | 1.7× | 0.1 | −0.1 | 0.1 | 0.4 | 0.2 | 0.2 | 0.1 |
| ❻ | scale (ViT-B, epochs=100) | ViT-B, epochs=300 | 7.2k | 3.0× | 1.6 | 0.5 | 0.4 | 1.7 | 1.3 | 0.5 | 1.0 |
| | | ViT-L, epochs=600 | 3k | 15× | 2.5 | 0.4 | 0.5 | 2.8 | 0.9 | 0.1 | 1.2 |

## 4 Ablation Analysis

### 4.1 CROMA Design

**Ablation Setup.** We ablate the CROMA design by pretraining CROMA-B models for 100 epochs unless stated otherwise. For each ablation, we set the maximum batch size that can fit into 640 GB of VRAM (with bfloat16 precision); adjusting the batch size results in better comparisons between approaches. For classification on BigEarthNet, we linear probe frozen unimodal and multimodal representations, i.e., $\text{Linear}(\mathcal{R}_R)$, $\text{Linear}(\mathcal{R}_O)$, $\text{Linear}(\mathcal{R}_{RO})$. For segmentation on DW-Expert-120, we linear probe frozen patch encodings, i.e., $\text{Linear}(\mathcal{E}_R)$, $\text{Linear}(\mathcal{E}_O)$, $\text{Linear}(\mathcal{E}_{RO})$. For BigEarthNet, we train on the same 10% split as §3.1, but report results on the combined validation and test sets (236,130 samples). For DW-Expert-120, we select $120 \times 120$ images (CROMA's default image size) from Dynamic World's [88] original Sentinel-2 images and match them, in space and time, with Sentinel-1 images—forming a high-quality multimodal segmentation dataset. DW-Expert-120 consists of 10,200 train and 45,367 validation samples.

**Ablation Results.** ❶ Removing either self-supervised objective significantly degrades accuracy on both classification and segmentation tasks. This justifies a fundamental design choice, i.e., combining contrastive and reconstruction approaches for aligned multimodal data. ❷ Although our primary motivation for 2D-ALiBi is to enable input size extrapolation (see §4.2), we find it also improves linear probing accuracy over a SoTA RPE method for ViTs, even when no extrapolation occurs. Averaged across six evaluations, 2D-ALiBi *without* X-ALiBi outperforms a Position Encoding Generator (PEG, [90]) by 0.5% and 2D-sinusoidal embeddings by 1.5%. (We adapt PEG to MAE-style masking by zero-filling the masked-out patches during pretraining, inspired by [91]). Leveraging X-ALiBi with 2D-ALiBi further improves average performance by 0.2% (Table 6)—particularly improving multimodal performance, which is our motivation for X-ALiBi. We believe there are two reasons for 2D-ALiBi's superior performance—evidence for both is provided in Appendix §A.2. First, 2D-ALiBi learns rotation-invariant representations despite not being trained with rotation-invariant objectives; this is a desirable property of satellite imagery that likely improves classification performance. Second, 2D-ALiBi prevents patch-wise representational collapse (i.e., the representations of patches within an image become similar, losing local information) often observed with contrastive objectives [47]; preserving patch-wise diversity likely improves segmentation performance. ❸ Lower mask ratios hurt classification and help segmentation but increase costs. For both 50% and 75% mask ratios, independently masking our modalities (i.e., optical and radar samples are masked differently) slightly outperforms shared masking. Although 50% independent masking outperforms 75% masking by 0.1%, we select the higher mask ratio because it offers a 1.8× speedup. ❹ A multimodal target (14 channels) outperforms an optical-only target (12 channels) by 0.2%, which outperforms a radar-only

target (2 channels) by $0.3\%$ (Table 6); this implies that reconstructing more information leads to more general representations. A multimodal target is especially important for learning rich multimodal representations. ❺ A larger decoder improves segmentation but costs $1.7\times$ more. MAE showed that a deep decoder improves linear probing accuracy [31], but CROMA is very robust to decoder size. Therefore, we select the most efficient, i.e., a 1-layer, 512-d decoder. ❻ We design CROMA by pretraining ViT-B models for 100 epochs and consider linear probing performance and cost. Thus, our design may not be optimal when scaling up. Nevertheless, scaling to more epochs and a larger model improves representations, especially radar-only representations.

In Appendix §A.1 we also experiment with—all showing minimal or negative impacts—task weights, more decoder sizes, VICReg [92], mean squared error loss between cross-modal patch encodings (inspired by [93]), InfoNCE loss between cross-modal patch encodings (inspired by [94]), hard-negative mixing [95], and lower-masked tuning at the end of pretraining [68].

## 4.2 Extrapolating Representations to Larger Images

**Extrapolation Setup.** For three CROMA-B models (2D-sinusoidal, PEG [90], and 2D-ALiBi) pretrained for 100 epochs, we finetune all parameters (along with a linear head) on Sentinel-2 images from DW-Expert-120 ($120 \times 120$ pixels). Then, we directly evaluate the models on images of varying sizes to test their ability to extrapolate at test-time. We create validation sets with different image sizes by cropping from the original $510 \times 510$ images in Dynamic World [88]. Regardless of image size, we retain the 10 m per pixel spatial resolution on which the models were trained—extrapolating to smaller or larger geographic areas at test-time. For 2D-sinusoidal embeddings, we also evaluate an embedding interpolation algorithm often used when applying ViTs on larger images [31, 26, 48, 96].

**Extrapolation Results.** 2D-ALiBi outperforms PEG by $1.7\%$ on $504 \times 504$ pixel images (Table 7). Amazingly, we only see a $0.7\%$ drop in mIoU when testing on areas $17.64\times$ larger than those on which our model was trained—effectively generalizing from 225 to 3,969 patches per image. We achieve this by extending ALiBi [66] to 2D inputs by penalizing attention scores based on the Euclidean distance between query-key pairs. We may achieve even better results by encoding directions in a subset of attention heads or learning scalars [97]; we leave these investigations to future work. We believe X- and 2D-ALiBi have tremendous potential beyond our CROMA framework, for instance, by extending these methods to additional modalities, viewing angles, or 3D data.

Table 7: Segmentation results (mIoU) of models trained on $120 \times 120$ resolution and evaluated on various resolutions.

| | Train Res. | | | | | | |
|---|---|---|---|---|---|---|---|
| Method / Test Res. | 48 | 96 | 120 | 224 | 384 | 448 | 504 |
| 2D-Sinusoidal | 54.4 | 58.5 | 59.2 | 38.2 | 24.9 | 21.8 | 19.5 |
| 2D-Sinusoidal w/ interp. | 50.7 | 58.6 | 59.2 | 58.3 | 56.2 | 55.5 | 54.9 |
| PEG [90] | 55.1 | 58.0 | 58.3 | 58.5 | 57.7 | 57.4 | 57.1 |
| **2D-ALiBi** | **56.3** | **59.0** | **59.3** | **59.5** | **59.1** | **59.0** | **58.8** |

## 5 Related Work

**Remote Sensing Representations.** Deep learning for remote sensing has been an active research area for many years. Researchers have leveraged self-supervised learning frameworks in the last few years to learn representations of remote sensing data that can be widely leveraged across societally important downstream tasks. In designing contrastive pretext tasks, remote sensing researchers built positive pairs by leveraging spatial [98, 99, 100, 101, 102, 103, 104, 105, 106, 107, 108], temporal [24, 25, 109], spectral [110], or cross-modal [111, 112, 113, 114, 115, 116] information. In designing reconstructive pretext tasks, researchers held-out spectral bands [117], pixels [118, 119, 120, 121], and resolutions [27]. However, these studies—including the concurrent Scale-MAE [27] and influential studies tile2vec [108], GASSL [24], and SeCo [25]—were either performed at smaller scales or only included wavelengths from the visible (red, green, and blue) spectrum; this limits their utility on downstream applications since wavelengths from the non-visible spectrum contain information *critical* to many remote sensing tasks [53, 122, 123, 124]. For example, the ability to measure the reflectance of objects in the near-infrared portion of the wavelength is extremely valuable in applications related to vegetation identification [125], health and productivity [126], as well as identifying water bodies [127], soil moisture [128], and vegetation water content [129].

**Relative Position Encoding for ViTs.** SoTA transformers in natural language processing use RPE [130, 131, 132], but fixed 2D-sinusoidal embeddings still predominate ViTs. The improved inductive

bias of RPE over absolute position encoding can offer improved performance and, sometimes, the ability to extrapolate to larger images at test-time, i.e., directly applying a model trained on one image size to another without further training. This extrapolation ability would add considerable value to remote sensing foundation models since image sizes vary widely—images are often cropped from large scenes down to smaller sizes chosen idiosyncratically. Positional Encoding Generator (PEG, [90]) is a SoTA RPE method that uses a convolution between ViT layers; PEG was demonstrated to significantly outperform other RPE methods when tested on larger images. iRPE [133] is another, more complex, RPE method for ViTs; however, it demonstrates no extrapolation ability. We found no prior work that leverages RPE in cross-attention.

## 6 Conclusion

We propose a novel framework for learning unimodal and multimodal representations for Earth Observation by jointly leveraging contrastive and reconstruction self-supervised objectives. We extend a SoTA position encoding method for 1D sequences to 2D inputs and cross-attention; to the best of our knowledge, this is the first time explicit position encoding has been leveraged in cross-attention. These strategies allow our models to extrapolate to larger images at test-time and improve performance on *both* unimodal and multimodal data. We *extensively* evaluate our pretrained models on diverse tasks and methods, outperforming the previous SoTA. Although our method is designed for multimodal satellite imagery, it can be leveraged on other applications that offer spatially aligned multimodal data, for instance, medical imaging or autonomous vehicles.

The main limitation of our work is our focus on static-in-time Sentinel-1 & 2 data; in the future, we will explore other sensors that offer higher spatial or spectral resolutions and time-series data. Despite this limitation, Sentinel-1 & 2 are the most widely used sources for satellite imagery in research—making our models an incredible resource for the remote sensing community and users of remote sensing-derived products, for instance, geographers, economists, or environmental scientists.

## 7 Acknowledgements

This work was made possible with compute provided by Cyxtera Technologies and the NVIDIA Academic Hardware Grant Program. Anthony thanks the Vector Institute for their financial support during his master's program.

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

# A Appendix

Code and pretrained models: `https://github.com/antofuller/CROMA`

## A.1 Other Pretraining Experiments

In this section, we experiment with additional CROMA settings. We use the same experimental conditions as §4.1 of our paper; i.e., we linear probe representations on BigEarthNet [76] (reporting mAP on the combined validation and test sets) and patch encodings on DW-Expert-120 [88] (reporting mIoU on the validation set). We use the linear probing hyper-parameters listed in §A.4.2 of this Appendix.

Table 8: Linear probing results on radar-only ("R"), optical-only ("O"), and joint radar-optical ("RO") inputs. Across all experiments, we use 2D-ALiBi with X-ALiBi, 75% shared masking, ViT-B backbones, and 100 pretraining epochs.

| Cross-Modal Image Obj. | Cross-Modal Patch Obj. | Decoder Depth, Dim | Obj. Weights $\lambda_{\mathrm{Con}}, \lambda_{MAE}$ | HN Mixing $(1024, 0, n)$ | Cost | Classification (mAP) | | | Segmentation (mIoU) | | |
|---|---|---|---|---|---|---|---|---|---|---|---|
| | | | | | | R | O | RO | R | O | RO |
| InfoNCE | MSE | 1, 512 | 1, 1 | ✗ | 1× | 77.4 | 83.9 | 84.3 | 40.5 | 56.0 | 56.7 |
| InfoNCE | MSE | 1, 768 | 1, 1 | ✗ | 1× | 77.4 | 83.9 | 84.3 | 40.7 | 56.1 | 56.6 |
| InfoNCE | MSE | 3, 512 | 1, 1 | ✗ | 1.2× | 77.5 | 83.9 | 84.4 | 40.5 | 56.3 | 57.1 |
| InfoNCE | MSE | 3, 768 | 1, 1 | ✗ | 1.3× | 77.5 | 83.9 | 84.4 | 40.7 | 56.2 | 56.6 |
| InfoNCE | MSE | 6, 512 | 1, 1 | ✗ | 1.4× | 77.5 | 83.9 | 84.4 | 40.3 | 56.0 | 56.7 |
| InfoNCE | MSE | 6, 768 | 1, 1 | ✗ | 1.6× | 77.6 | 83.8 | 84.5 | 40.6 | 56.2 | 56.7 |
| InfoNCE | ✗ | 1, 512 | 1, 1 | ✗ | 1× | 77.4 | 84.0 | 84.5 | 40.8 | 56.1 | 56.4 |
| InfoNCE | ✗ | 1, 768 | 1, 1 | ✗ | 1× | 77.5 | 84.2 | 84.5 | 40.8 | 56.1 | 56.2 |
| InfoNCE | ✗ | 3, 512 | 1, 1 | ✗ | 1.2× | 77.6 | 84.1 | 84.5 | 40.8 | 56.2 | 56.7 |
| InfoNCE | ✗ | 3, 768 | 1, 1 | ✗ | 1.3× | 77.0 | 83.9 | 84.5 | 40.6 | 56.1 | 56.5 |
| InfoNCE | ✗ | 6, 512 | 1, 1 | ✗ | 1.4× | 77.3 | 84.1 | 84.5 | 40.8 | 56.1 | 56.5 |
| InfoNCE | ✗ | 6, 768 | 1, 1 | ✗ | 1.6× | 77.5 | 84.1 | 84.6 | 40.6 | 56.5 | 56.8 |
| InfoNCE | InfoNCE | 3, 512 | 1, 1 | ✗ | 2.2× | 72.8 | 80.9 | 82.4 | 39.0 | 55.1 | 55.2 |
| InfoNCE | ✗ | 1, 512 | 1, 2 | ✗ | 1× | 77.5 | 84.3 | 84.2 | 40.7 | 55.9 | 56.2 |
| InfoNCE | ✗ | 1, 512 | 1, 4 | ✗ | 1× | 77.5 | 84.3 | 84.1 | 40.6 | 55.4 | 56.0 |
| InfoNCE | ✗ | 1, 512 | 2, 1 | ✗ | 1× | 77.5 | 84.1 | 84.5 | 40.4 | 55.9 | 56.3 |
| InfoNCE | ✗ | 1, 512 | 4, 1 | ✗ | 1× | 77.6 | 83.9 | 84.5 | 40.7 | 55.8 | 56.8 |
| InfoNCE | ✗ | 1, 512 | 1, 1 | 128 | 1× | 73.6 | 81.6 | 83.0 | 38.0 | 53.2 | 55.0 |
| InfoNCE | ✗ | 1, 512 | 1, 1 | 256 | 1× | 73.0 | 81.0 | 82.8 | 37.8 | 52.9 | 54.7 |
| InfoNCE | ✗ | 1, 512 | 1, 1 | 512 | 1× | 72.5 | 80.2 | 82.4 | 37.6 | 52.6 | 54.4 |
| VICReg | MSE | 1, 768 | 1, 1 | ✗ | 1.1× | 70.7 | 78.7 | 83.3 | 40.0 | 55.5 | 55.1 |

**Self-supervised Objectives.** Inspired by the local objective of VICRegL [93], we experiment with a mean squared error (MSE) objective between cross-modal patch encodings, i.e., $\mathcal{L}_{local}=\mathrm{MSE}(\mathcal{E}_{\mathrm{R}}, \mathcal{E}_{\mathrm{O}})$. This attracts patch encodings if they match locations, i.e., if they represent the same $80\,\mathrm{m} \times 80\,\mathrm{m}$ square on the ground. We find this does not improve representations. Next, we experiment with the VICReg [92] objective (calculating VICReg statistics based on a batch size of $800$) between cross-modal image representations, i.e., $\mathcal{R}_{\mathrm{R}}$ and $\mathcal{R}_{\mathrm{O}}$; we find it underperforms InfoNCE [28]. Finally, we experiment with the InfoNCE objective between cross-modal patch encodings; positive pairs are encodings that match locations across modalities, and negative pairs are all other encodings from the matched sample and encodings from all other samples in the batch. This does not improve representations and slows pretraining by $2.2\times$ (Table 8).

**Objective Weights.** We find that weighting the contrastive loss term or MAE [31] loss term does not uniformly improve representations; hence, we select equal weights.

**Hard Negatives.** We find that hard-negative mixing [95] ($N$=1024, $s$=0, $s'$=$n$, $\beta$=0.5, with $n$ of 128, 256, or 512) degrades performance when used in our framework.

**Decoder Sizes.** At least in these experiments, CROMA is not sensitive to the decoder size; a tiny decoder with a 1-layer, $512$-d transformer performs similarly to a much larger 6-layer, $768$-d transformer.

**Position Encoding with Shared Masking.** We find that using 2D-sinusoidal embeddings or PEG [90] with *shared* masking performs poorly. These two methods of position encoding store positional information in the internal representations, which can help solve the contrastive objective if both modalities share masks; 2D-ALiBi instead stores positional information in the attention matrix, which may pre-

Table 9: Linear probing results with *shared* 75% masking, ViT-B, 100 epochs.

| Method | Classification mAP | | | Segmentation mIoU | | |
|---|---|---|---|---|---|---|
| | R | O | RO | R | O | RO |
| PEG [90] | 67.9 | 75.9 | 79.0 | 32.6 | 49.8 | 51.0 |
| 2D-Sin. | 69.4 | 75.6 | 79.8 | 29.0 | 44.1 | 50.7 |

vent this from occurring. In our paper (Table 6), we show that 2D-sinusoidal or PEG can perform well in our framework if modalities are masked independently, although 2D-ALiBi still outperforms these approaches.

**Lower Masked Tuning.** FLIP [68] performs contrastive learning using the representations of masked-out samples; after this masked pretraining, it leverages *unmasked* tuning to increase accuracy by 1.3% on zero-shot ImageNet-1K. Unmasked tuning continues FLIP pretraining by performing contrastive learning using the representations of unmasked samples to reduce the distribution gap between pretraining and inference [68]. We cannot perform fully unmasked

Table 10: Lower masked tuning for 5 epochs after pretraining CROMA-L.

| Mask Ratio | Classification mAP | | | Segmentation mIoU | | |
|---|---|---|---|---|---|---|
| | R | O | RO | R | O | RO |
| 10% | 80.8 | 84.7 | 84.7 | 43.8 | 56.8 | 56.6 |
| 25% | 80.8 | 84.7 | 84.8 | 43.9 | 56.8 | 56.6 |
| 50% | 80.8 | 84.8 | 85.0 | 43.9 | 56.8 | 56.6 |

tuning because we must mask patches for our reconstruction objective. However, we can lower our mask ratio and perform *lower* masked tuning. Following FLIP, initializing parameters with our pretrained CROMA-L model, we train for 5 additional epochs using a base learning rate of 8e-8, warmup over the first epoch, and cooldown for 4 epochs using a cosine decay schedule. We explore mask ratios $\{10\%, 25\%, 50\%\}$ and find that lower masked tuning does not improve linear probing accuracy for CROMA (Table 10).

## A.2 Two Reasons for 2D-ALiBi's Performance

Our primary reason for introducing 2D-ALiBi is to enable the test-time extrapolation demonstrated in §4.2. But even when no extrapolation occurs—training and testing on the same image size—2D-ALiBi outperforms both the most commonly used position encoding method (2D-sinusoidal) and a SoTA relative position encoding method (PEG [90]). We believe 2D-ALiBi outperforms these two methods for two reasons.

Table 11: Cosine similarity between the representations of images and the representations of transformed versions of the same images. Higher similarity means the representations are less influenced by the image transformation.

| Method | Transformation (cosine similarity) | | | | |
|---|---|---|---|---|---|
| | H-Flip | V-Flip | rotate(90°) | rotate(180°) | rotate(270°) |
| 2D-ALiBi + X-ALiBi | 0.992 | 0.992 | 0.992 | 0.992 | 0.992 |
| 2D-ALiBi | 0.992 | 0.992 | 0.992 | 0.992 | 0.992 |
| PEG [90] | 0.992 | 0.992 | 0.988 | 0.992 | 0.988 |
| 2D-sinusoidal | 0.641 | 0.629 | 0.523 | 0.559 | 0.523 |

❶ **2D-ALiBi learns image representations invariant to rotating and flipping**. Learning representations that are invariant to certain transformations is often desirable. In CROMA, the contrastive objective between optical and radar data encourages sensor-invariant representations. Invariances to flipping and rotating are desirable properties of the representations of satellite imagery. However, CROMA's pretraining objectives do not explicitly encourage these invariances. To investigate if CROMA's optical representations (i.e., $\mathcal{R}_O$) are invariant to flipping and rotating, we produce representations of 5,000 optical images from DW-Expert-120 for four position encoding strategies—2D-ALiBi, 2D-ALiBi with X-ALiBi, PEG, and 2D-sinusoidal. Then, for each model, we produce representations of the same samples but transformed. We measure the cosine similarity between the original and transformed images' representations. When used in our CROMA framework, 2D-ALiBi learns representations invariant to flipping and rotating (average cosine similarity of 0.992, Table 11). Notably, 2D-sinusoidal learns representations that are not invariant to flipping and rotating

(average cosine similarity of 0.575, Table 11); we believe this contributes to the poor performance of 2D-sinusoidal embeddings on image classification. PEG performs similarly to 2D-ALiBi on image classification and also learns representations invariant to flipping and rotating.

Table 12: (Middle column) Cosine similarity between patch encodings at different locations within an image, averaged across 5,000 images. (Right column) Cross entropy loss of an MLP probe trained to predict patch locations given patch encodings.

| Method | Patch Encoding Similarity (cosine similarity) | Patch Encoding Position Probing (cross entropy loss) |
|---|---|---|
| 2D-ALiBi + X-ALiBi | 0.546 | 4.43 |
| 2D-ALiBi | 0.582 | 4.41 |
| PEG [90] | 0.701 | 4.13 |
| 2D-sinusoidal | 0.493 | 0.00 |

❷ **2D-ALiBi learns patch representations that retain local information**. [47] show that models trained with contrastive objectives can lose significant local information at deeper ViT layers that harm performance on dense prediction tasks. They show that the cosine similarity between the representations of patches at different locations within an image becomes high, indicating a patch-wise representational collapse. At every ViT layer, 2D-ALiBi injects a local bias in each attention head, for each patch location. To investigate if 2D-ALiBi can successfully prevent this collapse from occurring, we measure the cosine similarity between different patch encodings of the same sample and take the average across 5,000 optical images. We find that 2D-ALiBi learns to represent patches with greater spatial diversity than PEG, and X-ALiBi further improves diversity (Table 12). Interestingly, 2D-sinusoidal learns the most diverse patch representations. We also train MLP probes on patch encodings to classify the patch location; this measures the amount of positional information represented in patch encodings. We find that the patch encodings of models that use 2D-sinusoidal embeddings fully specify the location of the patch within the image (Table 12); this is an undesirable property of patch encodings, which should represent the *content* of the patch, rather than its location.

## A.3 Pretraining Details

### A.3.1 Data

We use the SSL4EO dataset [70], which consists of Sentinel-1 & 2 imagery acquired at 250K locations around the world; each location (a 2.64 km × 2.64 km square) is imaged four times, spread out over a year. We use these 1 million samples of 264 × 264 pixels for pretraining. Please see the SSL4EO paper [70] for more details.

### A.3.2 Implementation

We use an NVIDIA DGX server (8×A100-80GB), the maximum batch size that can fit into 640 GB of VRAM (7,200 for our default ViT-B), bfloat16 precision, a base learning rate of 4e-6, warmup for 5% of the total epochs, and cooldown via a cosine decay schedule. We use the same normalization procedure as SatMAE [26]. For data augmentation, we randomly crop 60-180 pixel squares from the original 264 × 264 pixels and resize the crops to 120 × 120 pixels (our default image size). We also perform vertical and horizontal flipping, 90-degree rotations, and mixup=0.3. Crucially, we apply these transformations identically to both modalities; if we applied them to each modality independently, our spatial alignment would break. We use the AdamW optimizer with $\beta_1$=0.9 and $\beta_2$=0.999, and a weight decay of 0.01.

## A.4 Evaluation Details

The evaluation of foundation models for Earth Observation is less mature than in other fields. We do our best to re-use the experimental conditions of the SoTA, i.e., SatMAE [26], and improve upon them where possible. One such condition is to report results from a held-out validation set; precisely, the best validation performance measured after each finetuning epoch is reported. No test sets are used. To enable fair comparisons with prior work, we copy this approach. In trying to improve the evaluation of foundation models for Earth Observation, we detail our approach in this Appendix, share code and preprocessed datasets, re-evaluate all near-SoTA models under identical

conditions, and evaluate models in more ways than prior work (i.e., linear and nonlinear probing, $k$NN classification, and $K$-means clustering).

We initialize parameters from publicly shared pretrained weights, evaluating all models ourselves under identical conditions. Although this process is laborious, we believe it significantly improves the value of our paper; several prior studies have often evaluated their models in different ways, using different data splits that cannot be directly compared. When downloading pretrained weights, we use the latest weights that are publicly available. For instance, SatMAE [26] released improved versions of their multispectral ViT-B and ViT-L models, pretrained for 200 epochs, after their manuscript was accepted for publication (edited on *arxiv* on January $15^{th}$, 2023). We exclusively evaluate these improved models throughout our paper, ensuring we compare CROMA to the best models available. For multispectral benchmarks with 13 channels (with cirrus included), we simply drop the cirrus band for models pretrained without it.

### A.4.1 Data

**BigEarthNet.** [76] We use the same splits for training (10% of the complete training set) and evaluating (the entire validation set) as SatMAE [26] and SeCo [25]. However, we use the combined validation and test sets (236,130 samples) in our ablation studies to increase the reliability of our findings with minimal added cost. Images are $120 \times 120$ pixels.

**fMoW-Sentinel.** [26] Inspired by how the BigEarthNet benchmark is used (i.e., training on 10% of the complete training set of 354,200 samples), we create a 10% split of the complete fMoW-Sentinel training set of 712,874 samples. We share the IDs of the 10% of fMoW-Sentinel training samples that we randomly selected. We believe this smaller training set should be used in future work to reduce the costs of hyper-parameter searches—a *single* finetuning run of SatMAE on the complete training set requires 192 hours on a V100 GPU [26]. Following SatMAE, we use the full validation set for evaluation. Images vary in size; the mean height is 45 pixels, and the mean width is 60 pixels.

In our paper, we benchmark this new split. However, we report results obtained by our CROMA models on the complete training set in Table 13. Due to the costs of finetuning on the complete training set (712,874 samples), we decide to allocate our resources elsewhere and *not* perform any hyper-parameter tuning. Instead, we select hyper-parameters we believe to be reasonable and finetune CROMA-B and CROMA-L once. For finetuning, we use a base learning rate of 1e-5 and all other hyper-parameters from §A.4.2.

Table 13: fMoW-Sentinel results (top 1 accuracy) using the *complete* training set. * denotes results reported in SatMAE (updated on *arxiv* on January $15^{th}$, 2023).

| Method | Backbone | Finetuning | Linear Probing |
|--------|----------|------------|----------------|
| SatMAE | ViT-B | 62.65* | 37.40 |
| CROMA | ViT-B | 61.00 | 40.94 |
| SatMAE | ViT-L | 63.84* | 39.19 |
| CROMA | ViT-L | 63.59 | 41.96 |

**EuroSAT.** [77] We use the same training and validation sets as SatMAE. Images are $64 \times 64$ pixels.

**Canadian Cropland.** [78] We are the first to benchmark this dataset of Canadian agricultural croplands, consisting of 10 classes (barley, canola, corn, mixedwood, oats, orchard, pasture, potato, soybean, and spring wheat). We select this dataset because it is a large dataset that evaluates different capabilities from the other benchmarks that typically consider croplands as a single class. Following EuroSAT [77], the authors selected an image size of $64 \times 64$ pixels [78]; therefore, models evaluated on EuroSAT can be evaluated on Canadian Cropland with minimal modifications. We use the training set and combine their validation and test sets to form a single held-out set for evaluation. We share these complete training and validation sets. The performance (Table 1) and representation visualizations (Fig. 6 and 7 in this Appendix) indicate that the 10 classes present in this dataset are challenging to separate.

**DFC2020.** [87] This dataset is used for evaluation in diverse ways—both the choice of data split and image size. The original dataset comprises 6,114 samples of $256 \times 256$ pixels. These samples are typically split into two: a so-called "validation set" of 986 samples and a so-called "test set" of 5,128 samples. Some studies use the "validation set" for training and the "test set" for validation; others use the "test set" for training and the "validation set" for validation. Some studies use the full $256 \times 256$ pixels as inputs to their models, while others use smaller inputs. We select the split of 5,128 samples for training, which we divide into 46,152 images of $96 \times 96$ pixels—leaving us with the split of 986 samples for validation, which we divide into 8,874 images of $96 \times 96$ pixels. We select this final

resolution because it is the default image size of SatMAE, enabling a fair comparison to the SoTA. We share these complete training and validation sets.

**DW-Expert.** [88] The data collected by Dynamic World [88] is a new high-quality dataset annotated by experts with the help of auxiliary information. Thus, it should be used in the future when benchmarking models. Our work uses the expertly annotated data from Dynamic World, which we split into 20,422 train samples and 51,022 validation samples. All images are $96 \times 96$ pixels to enable a fair comparison with SatMAE. We share these complete training and validation sets. We also create a version of this dataset that consists of $120 \times 120$ pixel images (i.e., DW-Expert-120) that we only use for ablations because it is the default image size of CROMA.

**MARIDA.** [89] We use the training set and combine the validation and test sets to form a single held-out set for evaluation. Following our approach for DFC2020 and DW-Expert, we divide the original images into images of $96 \times 96$ pixels. Because it is a sparsely labeled dataset (i.e., only a fraction of pixels per image are labeled), we include images with at least one labeled pixel. We select this dataset because it evaluates different capabilities from the other semantic segmentation benchmarks. It consists of the following classes: marine debris, dense *Sargassum*, sparse *Sargassum*, natural organic material, ship, clouds, marine water, sediment-laden water, foam, turbid water, shallow water, waves, cloud shadows, wakes, and mixed water. We share these complete training and validation sets.

### A.4.2 Implementation

**Finetuning.** We select reasonable hyper-parameters that we use for all models and datasets unless otherwise stated and sweep across learning rates. This learning rate sweep is essential to creating fair evaluation conditions across models since each model is given the same search budget (in terms of finetuning runs, not compute hours), and different models have different optimal learning rates. Models pretrained with reconstruction approaches tend to require higher base learning rates during finetuning than models pretrained with contrastive learning. For instance, MAE [31] lists a base learning rate of 1e-3, FLIP [68] lists a base learning rate of 5e-5, CoCa [49] lists base learning rates from 1e-5 to 5e-4, depending on the downstream dataset.

No single learning rate would enable a fair comparison across all models and datasets. Therefore, we sweep learning rates across an extensive range {3e-5, 5e-5, 8e-5, 1e-4, 3e-4, 5e-4, 8e-4, 1e-3} and report the best single evaluation result obtained for each dataset; this sweep is performed for CROMA models and all other models. We convert these base learning rates to actual learning rates via the widely used linear scaling rule: $lr = base\_lr \times batch\_size/256$. We use the largest batch size that can fit on an A100-40GB GPU (using bfloat16 precision), the AdamW optimizer with $\beta_1$=0.9, $\beta_2$=0.999, and a weight decay of 0.01. We warmup for 5 epochs and cooldown for 30 epochs using a cosine decay schedule (other than EuroSAT, which we cooldown for 150 epochs); this follows SatMAE [26]. For classification tasks, we use mixup=0.8, cutmix=1.0, switch probability=0.5, label smoothing=0.1, vertical and horizontal flipping, and 90-degree rotations. We enlarge images to the default image size of the model we are finetuning (i.e., the image size on which the model was pretrained), with one exception. The default image size of SatMAE is $96 \times 96$; however, BigEarthNet images are $120 \times 120$ [76], requiring that we either crop BigEarthNet samples (losing information) or adapt SatMAE to larger images. We achieve better performance by adapting SatMAE to $120 \times 120$ images via the widely used position embedding interpolation algorithm than cropping BigEarthNet samples down to $96 \times 96$. This allowed us to achieve an mAP of 86.18 for SatMAE, a significant improvement over the 82.62 reported in the SatMAE paper. All other datasets use images of $96 \times 96$, or smaller—thus, there is no reason to use this technique for other datasets.

**Linear and Nonlinear Probing.** We encode each image without data augmentation then train linear and nonlinear probes on the frozen representations. Since each model only encodes each image once, we can sweep through a large range of learning rates ({1, 2, 3, 4, 5, 6, 7, 8, 9}e{-4, -3, -2}) very quickly. Unlike finetuning, we do not evaluate probes after every epoch; instead, we evaluate trained probes after all epochs are complete. We use a batch size of 1024, bfloat16 precision, the AdamW optimizer with $\beta_1$=0.9, $\beta_2$=0.999, and a weight decay of 0.01. We warmup for 5 epochs and cooldown for 100 epochs using a cosine decay schedule.

**Non-parametric $k$NN and $K$-means.** For $k$NN, we use the implementation from [27]. This consists of encoding all training and validation samples and then using the representations of validation samples as queries and training samples as keys to fetch training labels. These fetched training labels are used to classify validation samples. We use $k$=20, other values for $k$ (i.e., 10, 50) ranked

Table 14: CROMA vs SatMAE training and inference throughput on an A100-40GB GPU.

| Model | Backbone | Image Size | Train Imgs/s | Inference Imgs/s |
|---|---|---|---|---|
| SatMAE | ViT-B | 96×96 | 249.3 | 692.5 |
| CROMA | ViT-B | 96×96 | 1,079.1 | 2,957.7 |
| CROMA | ViT-B | 120×120 | 555.0 | 1,532.1 |
| SatMAE | ViT-L | 96×96 | 84.2 | 263.2 |
| CROMA | ViT-L | 96×96 | 389.1 | 1,168.2 |
| CROMA | ViT-L | 120×120 | 209.6 | 640.3 |

models in the same order as $k$=20. For $K$-means, we use the implementation from [79]. This consists of encoding all training and validation samples, then clustering training samples with $K$-means ($K$-means++ [134] initialization run 10 times). Then, we assign validation samples to clusters and assign clusters to classes via the Hungarian matching algorithm [135].

**Sparse Probing.** For each model and dataset, we rank the dimensions of representations $\mathcal{R}$ using the *mean difference* between classes—[86] shows that the mean difference performs on-par with more complex ranking methods and is simple to implement. Specifically, this is our procedure to sparsely probe the representations of CROMA for BigEarthNet's "beaches, dunes, sands" class: (i) compute the average representation, $\mathcal{R}_{\text{CROMA}}^{\text{Beaches}} \in \mathbb{R}^{768}$, of all samples in the BigEarthNet training set that contain the class "beaches, dunes, sands"; (ii) compute the average representation, $\mathcal{R}_{\text{CROMA}}^{\text{NoBeaches}} \in \mathbb{R}^{768}$, of all samples in BigEarthNet that *do not* contain the class "beaches, dunes, sands"; (iii) compute the difference between these averaged representations, $\mathcal{R}_{\text{diff}} \in \mathbb{R}^{768}$; (iv) rank all 768 dimensions in $\mathcal{R}_{\text{diff}}$ by the absolute value, i.e., the dimension with the greatest absolute difference between classes is ranked first; (v) train a separate linear probe to perform binary classification on the top $k$ dimensions, we sweep many values of $k$ between 1 and 768; (vi) using these trained probes, evaluate them on the BigEarthNet validation set. Again, this procedure is performed for all three ViT-B models (CROMA, SatMAE, and I-JEPA), all classification datasets (BigEarthNet, fMoW-Sentinel, Canadian Cropland, and EuroSAT), and all classes—these plots are displayed at the end of this Appendix §A.6. We also sparsely probe radar-only, $\mathcal{R}_{\text{R}}$, and radar-optical, $\mathcal{R}_{\text{RO}}$, representations for BigEarthNet, since we have access to Sentinel-1 samples.

**SatMAE Specifics.** SatMAE [26] divides spectral bands into three groups and outputs patch encodings for every group; thus, SatMAE outputs three patch encodings per patch location. To be as fair as possible to SatMAE, we explore four ways of merging these co-located patch encodings to perform segmentation: unnormalized spatial concatenation, normalized spatial concatenation, unnormalized spatial pooling, and normalized spatial pooling. We find unnormalized spatial concatenation (i.e., concatenating the patch encodings of co-located patches before the LayerNorm) performed best. Thus, we use the unnormalized spatially concatenated patch encodings for all segmentation datasets. Conversely, CROMA does not divide spectral bands into groups—resulting in $3\times$ shorter sequence lengths. The computation required to process a sequence of tokens with a transformer increases with increasing sequence lengths. This makes CROMA much more computationally efficient than SatMAE for a given ViT backbone and image size (Table 14). We also pretrain a SatMAE-B model for 300 epochs on the SSL4EO dataset. However, this model performs poorly, so we do not report these results; this experiment indicates that SatMAE's hyper-parameters may not transfer well to different pretraining datasets.

### A.5 Societal Impact

Since we pretrain our models on the SSL4EO dataset [70], our models may be biased towards the distribution from which SSL4EO data were sampled. Although SSL4EO samples are geographically diverse (please see Fig. 2 from the SSL4EO paper [70]), locations are sampled from areas surrounding human settlements. As a result, large geographic areas that are sparsely populated—for instance, the Amazon rainforest, the Sahara desert, and the Australian outback—are underrepresented. This could negatively impact the quality of representations in these locations and any decisions made on their basis.

Another distribution shift—this time, between finetuning and inference—is our primary concern. For example, finetuning a model on the imagery of one geography and then making predictions on the imagery of another geography creates a distribution shift. As a result, biases from the finetuning geography may be realized in the predictions made by the finetuned model. This is particularly

problematic when these predictions are used in decision-making, for instance, allocating poverty assistance. However, it is well-demonstrated that pretrained models are more robust to distribution shifts than models trained from scratch. Additionally, as we develop better foundation models for Earth Observation, we reduce the need for annotated data; this may allow practitioners to be more selective of the data they wish to leverage during finetuning.

We do not expect our pretrained models to be particularly valuable for military applications, as militaries likely have access to higher resolutions (spatially, spectrally, and temporally) than Sentinel-1 & 2 provide. However, our framework may be leveraged to pretrain models on higher-resolution imagery, which could be useful for military applications, although this is a risk of all novel learning algorithms.

### A.5.1 Compute

We approximate the computational resources we use for pretraining and finetuning (frozen representation evaluations are negligible in comparison). For pretraining, estimates are in A100-80GB GPU hours; for finetuning, estimates are in A100-40GB GPU hours. Please see Table 15.

Table 15: Estimated GPU hours used for developing and validating CROMA.

| Method | Backbone | Task | GPU Hours |
|---|---|---|---|
| radar↔optical [71] | ResNet50 | Classification Finetuning | 10 |
| radar↔optical [71] | Swin-T | Classification Finetuning | 25 |
| MAE [31, 70] | ViT-S | Classification Finetuning | 20 |
| DINO [72, 70] | ViT-S | Classification Finetuning | 20 |
| SatMAE [26] | ViT-B | Classification Finetuning | 75 |
| CROMA | ViT-B | Classification Finetuning | 35 |
| SatMAE [26] | ViT-L | Classification Finetuning | 215 |
| CROMA | ViT-L | Classification Finetuning | 90 |
| CROMA | ViT-B | Pretraining 300 epochs | 80 |
| CROMA | ViT-L | Pretraining 600 epochs | 380 |
| CROMA | ViT-B | Pretraining Ablations | 1,100 |

### A.6 Visualizations

We visualize representations and patch encodings using UMAP and t-SNE. For both segmentation datasets (DFC2020 [87] and DW-Expert [88]), we visualize patch encodings of 50,000 randomly sampled patches and use the most dominant class in a patch as its label. We also plot sparse probing results for binary classifiers trained on the top $k$ representation dimensions.

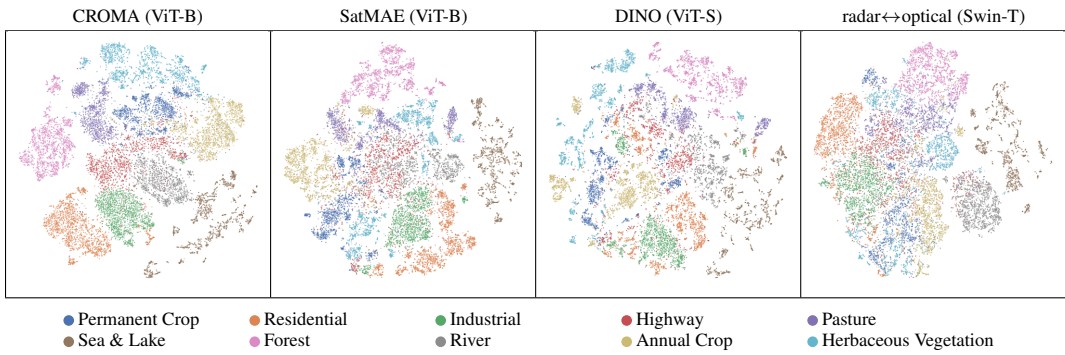

Figure 5: t-SNE plots of EuroSAT [77] representations.

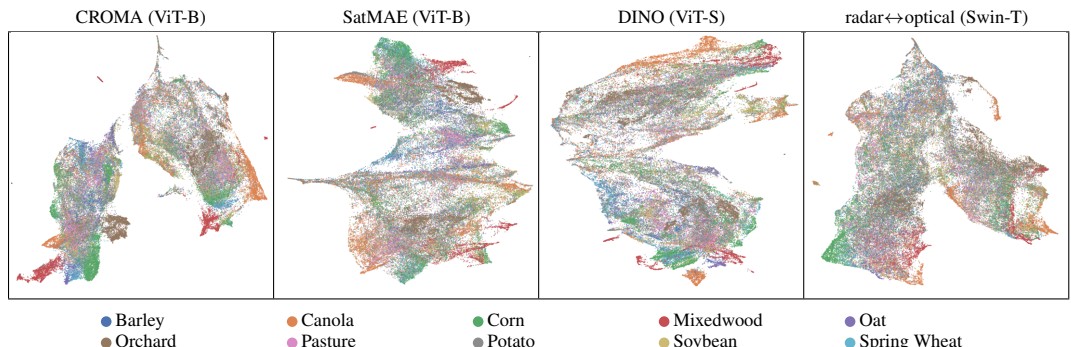

Figure 6: UMAP plots of Canadian Cropland [78] representations.

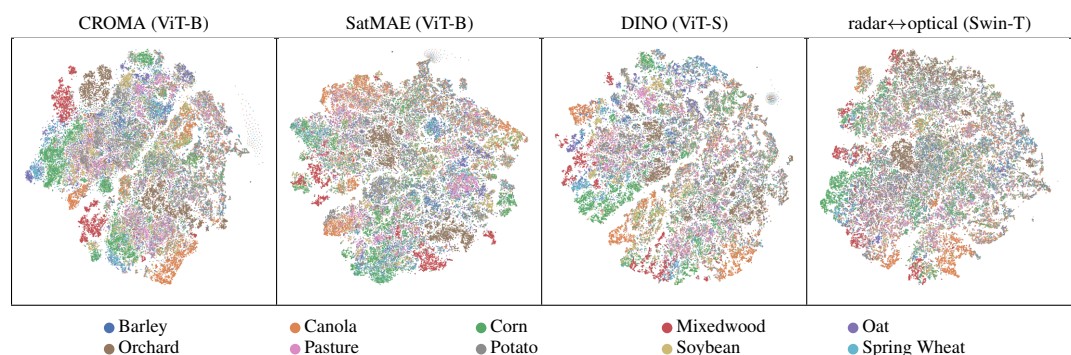

Figure 7: t-SNE plots of Canadian Cropland [78] representations.

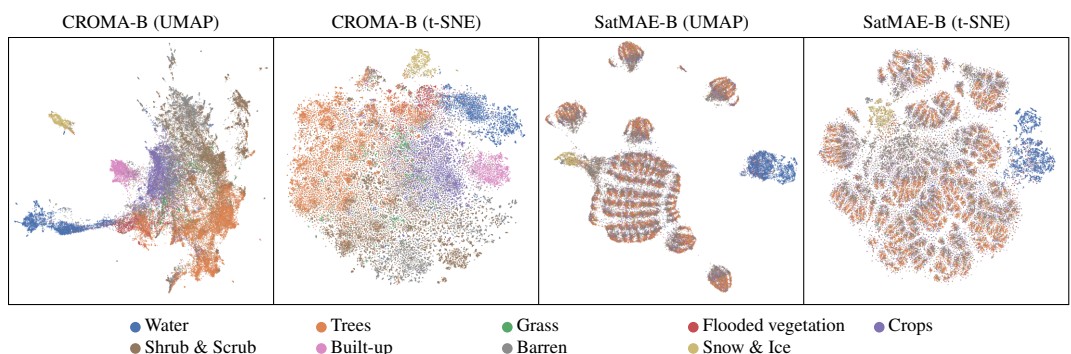

Figure 8: UMAP and t-SNE plots of DW-Expert [88] patch encodings.

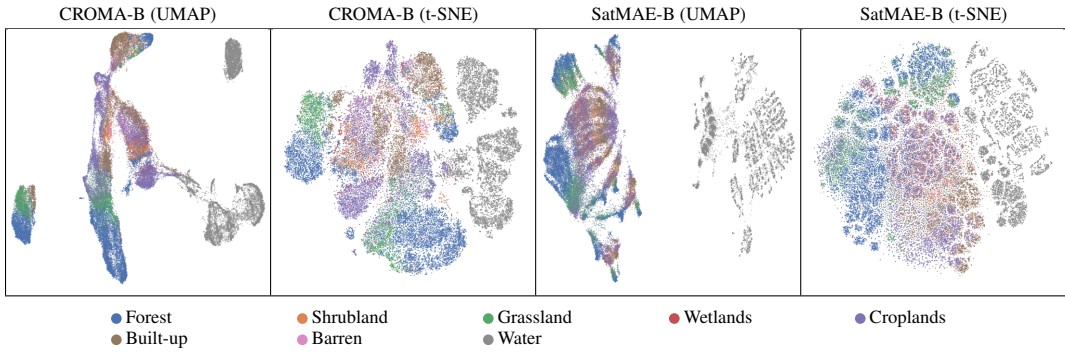

Figure 9: UMAP and t-SNE plots on DFC2020 [87] patch encodings.

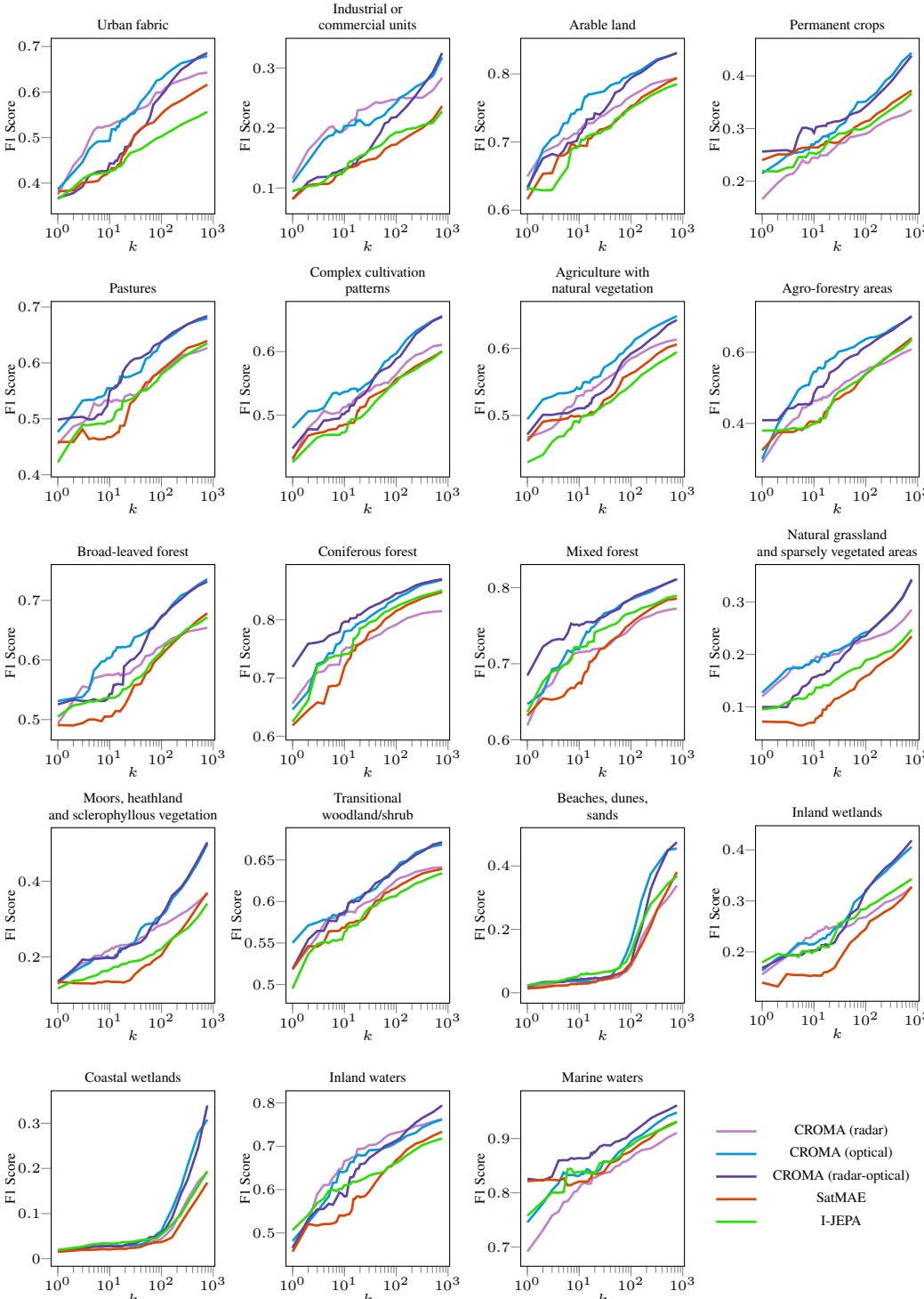

Figure 10: Sparse Probing all classes in BigEarthNet

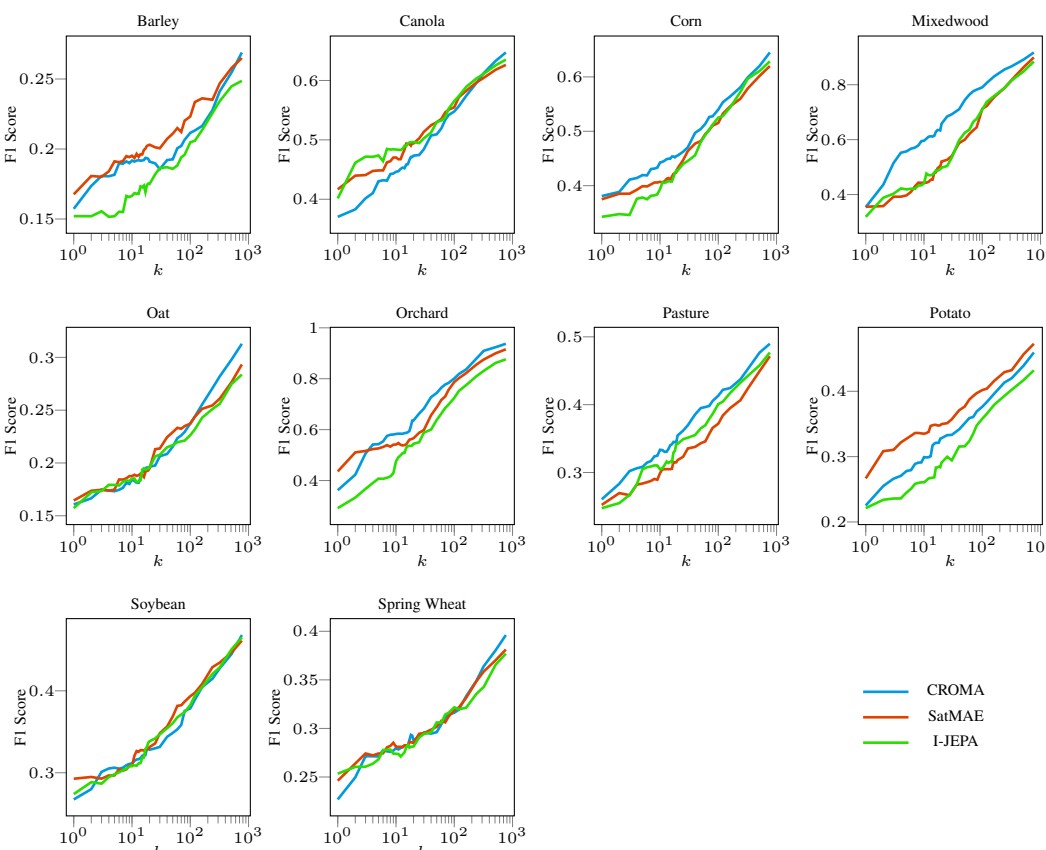

Figure 11: Sparse Probing all classes in Canadian Cropland

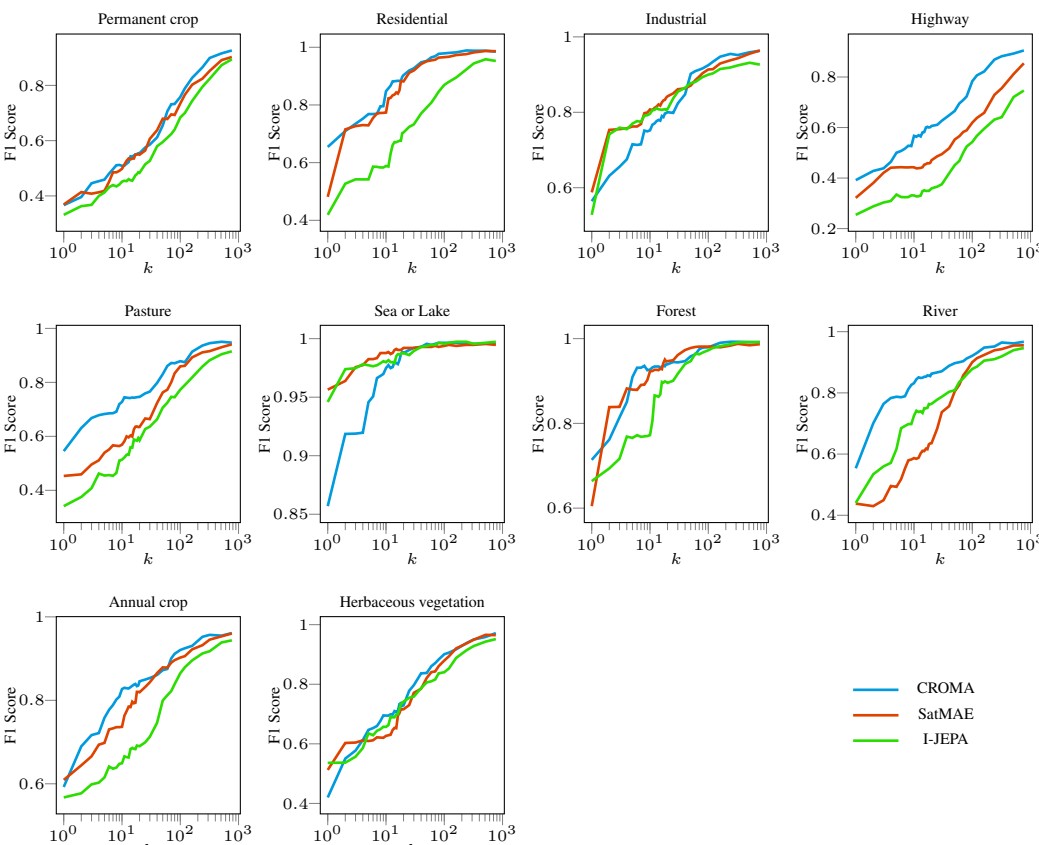

Figure 12: Sparse Probing all classes in EuroSAT

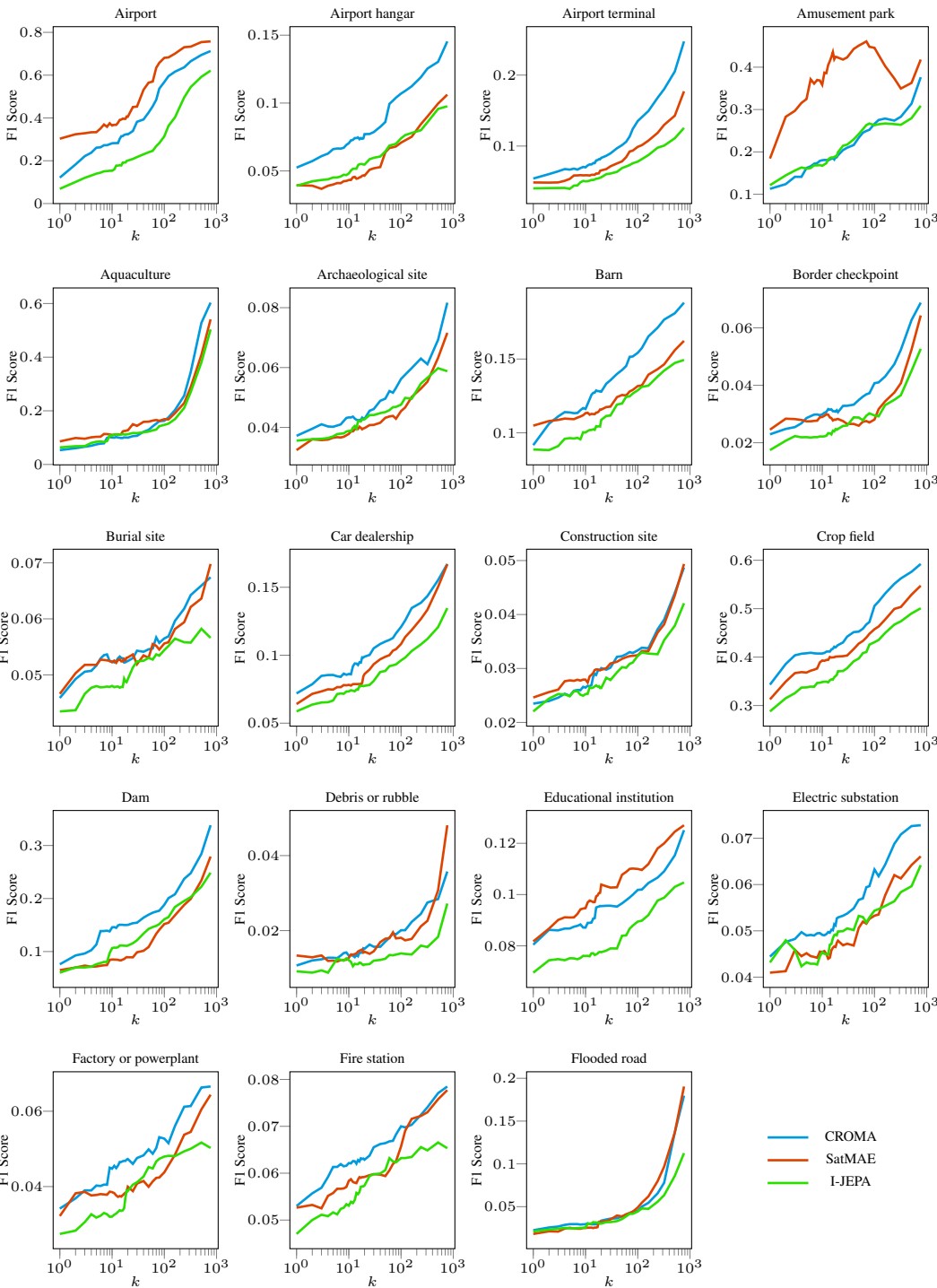

Figure 13: Sparse Probing all classes in fMoW-Sentinel (part i)

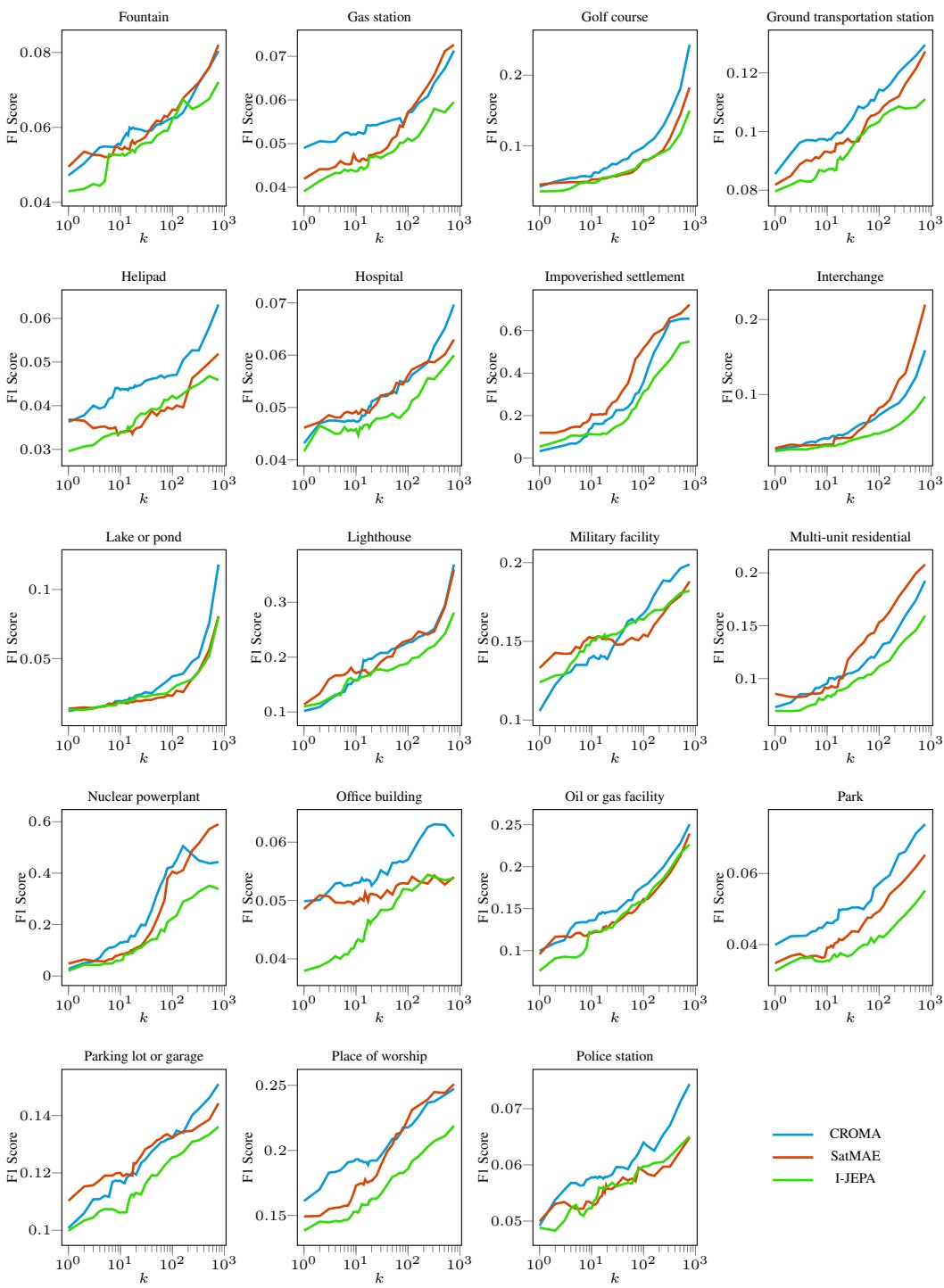

Figure 14: Sparse Probing all classes in fMoW-Sentinel (part ii)

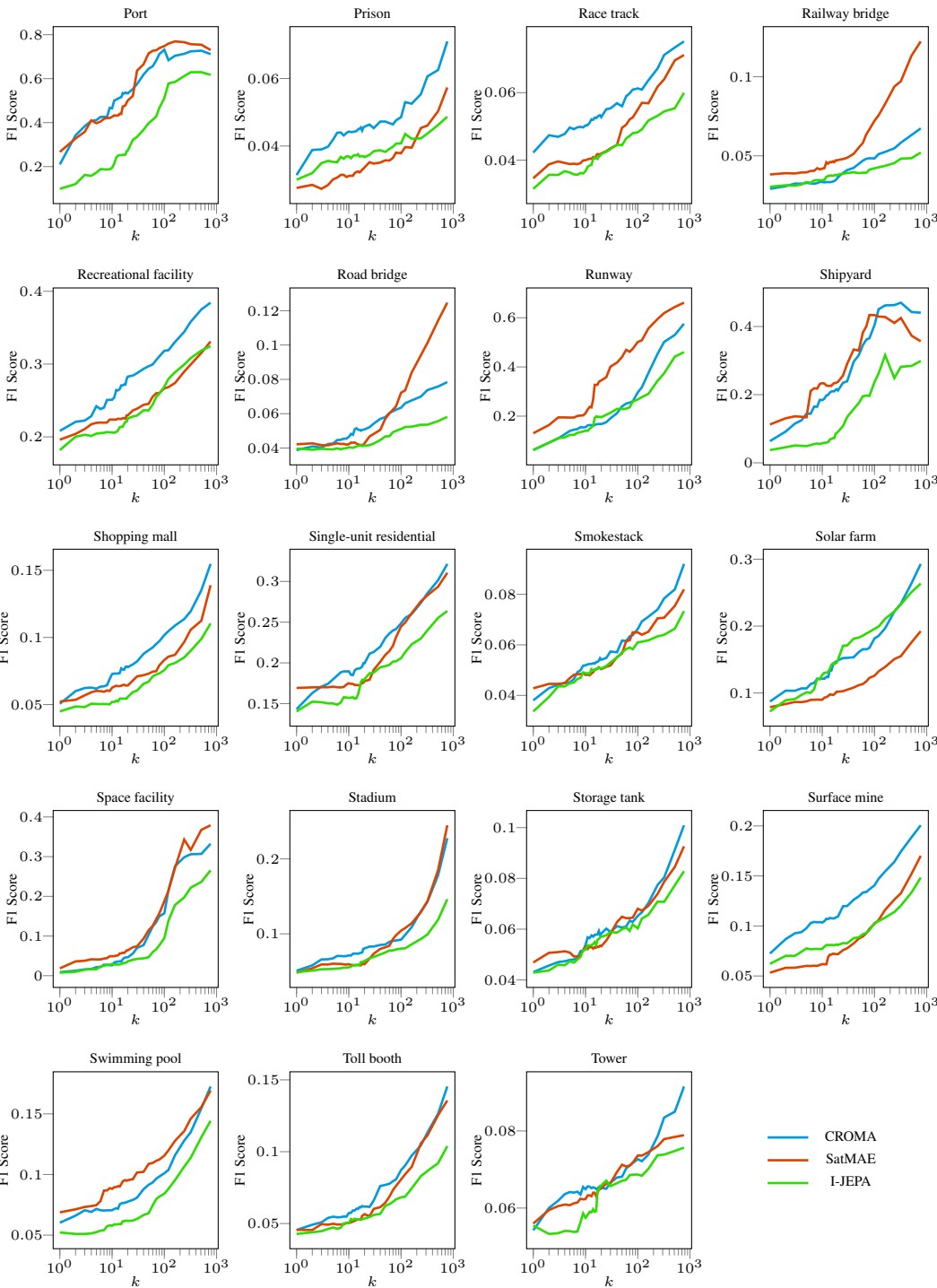

Figure 15: Sparse Probing all classes in fMoW-Sentinel (part iii)

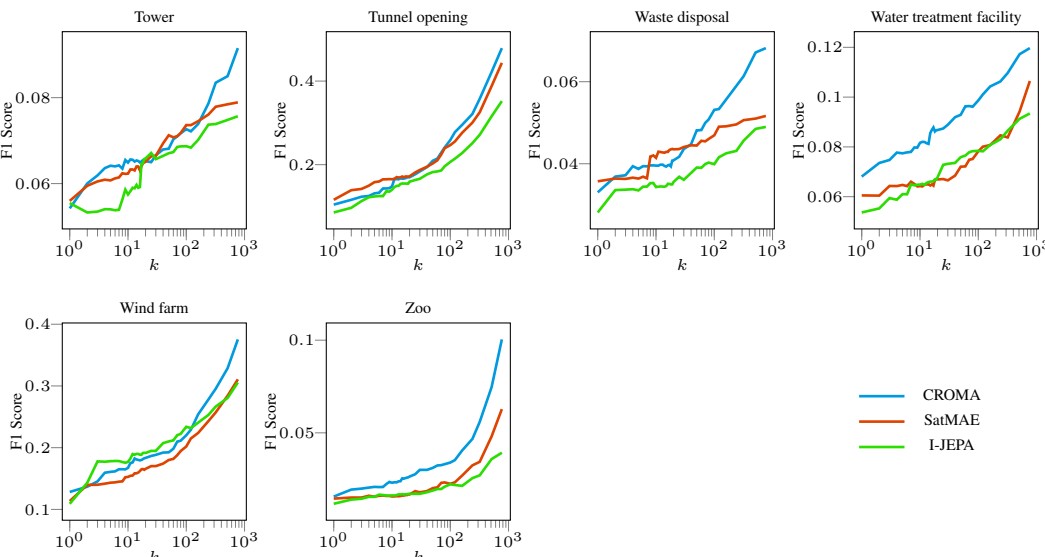

Figure 16: Sparse Probing all classes in fMoW-Sentinel (part iv)

