# OpenReview forum: "CROMA: Remote Sensing Representations with Contrastive Radar-Optical Masked Autoencoders"
_NeurIPS.cc/2023/Conference — NeurIPS 2023 poster_

### Official Review · Reviewer_FExC · 2023-06-29

**Soundness:** 4 excellent
**Presentation:** 4 excellent
**Contribution:** 3 good
**Rating:** 7
**Confidence:** 5

**Summary:**

The paper presents a new SSL representation learning framework remote sensing and earth observation data. The presented framework combines a contrastive objective with a reconstruction objective working on single or multi-modal inputs i.e. multispectral satellite data and synthetic aperture radar data. Cross-modal learning is done by cross-attention of both individually encoded modalities, which are fused and decoded by one lightweight decoder.

In a wide range of experiments, the authors are able to demonstrate the proposed approach capability to outperform baseline and current SOTA approaches.

Although this work displays rather a novel combination of already known approaches, I think it is interesting given the insightful adaptation of these approaches to the remote sensing and earth observation domain. I really enjoyed reading it. What I really like is the idea to use RPE as presented by the extension of ALiBi towards multispectral 2dim signals allowing to deal with different resolutions of satellite data. This particular characteristic of satellite data is very often neglected.

**Strengths:**

- (S1) The use of RPE with its 2d extension of ALiBi including X-ALiBi. I think this is interesting since it aims to tackle the multi-resolution nature of individual bands of remote sensing data.

- (S2) multi-modal representation is optional i.e. it performs well with only one modality if needed. This is in particular interesting for remote sensing scenarios such as natural disasters, where fast response is important but one of the two satellites is not available but will need a couple of days to fly over the target region.

- (S3) The wide range of experiments including multiple datasets and downstream tasks all able to demonstrate the outperformance of the presented approach.

- (S4) A broad set of ablation studies providing insights about the inter-working and contribution of each component of the proposed method.

**Weaknesses:**

- (W1) see (L1) under limitations.

**Questions:**

- (Q1) How would you extend to more than two modalities? I am asking since in remote sensing and earth observation there are very often multiple modalities / sensors available. How would you model the cross-attention?

**Limitations:**

- (L1) The paper does not show the presented approach being capable of generalizing to other problem domains beyond remote sensing or earth observation. I could imagine that there exist other problem domains where multiple sensors are available. Showing how this approach performs in such a scenario would strengthen this work.

---

> ### Author Rebuttal · Authors · 2023-08-10
>
> Thank you for your thoughtful comments on our work. We appreciate you recognizing the strengths of our work: (i) the introduction of X- and 2D- ALiBi, (ii) the optionally multimodal nature of CROMA, (iii) the extensive evaluation across methods, tasks, and datasets, and (iv) the thorough ablation. Although our paper is dense, we appreciate that the paper was really enjoyable to read. We will address two points individually:
>
> **Q**: *How would you model the cross-attention with more than two modalities?*
>
> **A**: Extending CROMA to more than two modalities is very exciting. Let’s imagine we are provided with higher-resolution RGB imagery spatially aligned with Sentinel-1 and 2 data. In this case, we could encode this high-res sample separately with a unimodal encoder, just like CROMA encodes radar and optical samples. Then a multimodal encoder could cross-attend to both sets of optical encodings (high-res and Sentinel-2) and bias the cross-attention matrix based on the distance between patches via X-ALiBi. In practice, this would mean repeating the X-ALiBi bias matrix along the key dimension, i.e., from a shape of (batch_size, heads, queries, keys) to (batch_size, heads, queries, 2*keys). If the patches between modalities do not perfectly align—for example, if the high-res sample used patches that were 16x16x3 at 1m spatial resolution—then we’d have to build another X-ALiBi matrix by calculating the relative distance between each query and key patch and concatenate this high-res bias matrix with the Sentinel-2 bias matrix along the key dimension. If there were 196 high-res patches, then the new X-ALiBi bias matrix would be of shape (batch_size, heads, queries, keys+196). In general, we believe that as long as the relative locations between cross-modal patches are known, then X-ALiBi can be leveraged. For example, imagine we have ground-level imagery with known coordinates. We could bias the cross-attention matrix based on the 2D distance between a ground-level representation and various patches in the satellite image. This idea could be extended to multiple ground-level images for a single satellite image. This research direction excites us—thank you for prompting this discussion.
>
> **Q**: *Validation on other domains with multiple sensors.*
>
> **A**: We agree that validating CROMA on other sensor data would strengthen our work. However, our paper is already quite dense. We look forward to seeing other groups apply CROMA to their applications. The potential for CROMA to be used broadly is one of our main reasons for targeting NeurIPS.

---

> > ### Comment · Reviewer_FExC · 2023-08-20
> > **Response**
> >
> > I would like to thank the authors for their time and the level of details provided to address my questions.
> >
> > In particular, thanks for the in-depth explanation of X-ALiBi's capability to be extended towards multiple modalities, not only coming from different (registered) sensors but also from different resolutions. This renders the presented approach to be more scalable wrt. to input modalities than published work in this area employing straightforward contrastive learning approaches for pre training.
> >
> > Given that my questions are answered and I think that this submission outlines a very interesting research direction to investigate, I decide to **increase** my rating.

---

### Official Review · Reviewer_vsHk · 2023-07-01

**Soundness:** 2 fair
**Presentation:** 1 poor
**Contribution:** 1 poor
**Rating:** 3
**Confidence:** 5

**Summary:**

This paper presents a CROMA, a framework that combines contrastive and reconstruction self-supervised objectives to learn rich unimodal and multi-modal representations. CROMA separately encodes masked-out multispectral optical and synthetic aperture radar samples and performs cross-modal contrastive learning. X- and 2D-ALiBi are also introduced to ensure the performance, which spatially biases the cross and self-attention matrices.

**Strengths:**

CROMA aims to address the multi-model learning problem in the remote sensing (RS) community, which is an important and hot topic. Also, many advanced techniques are adopted and combined properly to ensure the final results for different tasks. In sum, the proposed method is feasible.

**Weaknesses:**

However, due to the poor statements and organization, the main ideas of this work are hard to follow. Also, as mentioned above, CROMA combines some existing techniques to deal with its tasks. Thus, its novelty is limited for NIPS. Some detailed comments can be found in “Questions.”

**Questions:**

1.	The main contributions are not clear. Many multi-model learning models have been proposed, what are your advantages and own features compared with them?
2.	What is FFT? I cannot find its full name in this manuscript.
3.	There are three encoders in CROMA. What are the relations between them? What is the rationale behind the freed settings and parameters? How do you decide the input patch sizes?
4.	Why do you extend ALiBi to a 2D version? Please explain the necessity.
5.	How do you decide the MASK value (i.e., 75%)?


**Limitations:**

1.	Introduction is chaotic. There is no clear, logical flow between paragraphs, resulting in a disjointed content presentation.
2.	A meaningful literature review is missing. The authors only display many published literature. However, the relations between them and CROMA are not clear. Also, the inner relationships of the reviewed literature are confusing.
3.	The experimental settings are unclear, preventing the readers from simulating your method.
4.	The compared models are not enough, limiting the reliability of the results.
5.	The experimental results are discussed as shallow.

---

> ### Author Rebuttal · Authors · 2023-08-10
>
> Thank you for your thoughtful comments on our work. Please see our replies below:
>
> **Q**: *Clarification of main contributions.*
>
> **A**: Please see our comments to all authors that clarify our contributions.
>
> **Q**: *What is FFT?*
>
> **A**: The term FFT does not appear in our document; perhaps the reviewer is referring to FFN? FFN stands for feedforward network—this is the standard way of referring to the feedforward network in a transformer. We will define the term in our revision.
>
> **Q**: *Clarification on CROMA’s three encoders and choice of patch size.*
>
> **A**: CROMA uses three encoders, (i) a unimodal radar encoder, (ii) a unimodal optical encoder, and (iii) a multimodal (i.e., radar-optical) encoder. The unimodal encoders use a standard ViT-B architecture (other than position encoding) with 8x8 patches. ViTs often use 16x16x3 or 8x8x3 patches (HxWxC); since our optical samples have 12 channels, 16x16x12 patches would lose information when linearly projected to the width of the transformer (i.e., 768 for ViT-B or 1024 for ViT-L). Conversely, selecting patches that are too small (i.e., when the number of pixels per patch is far fewer than the width of the transformer) leads to inefficient models [136]. Thus, an 8x8 patch size is a reasonable choice for CROMA.
>
> The multimodal encoder in CROMA is a transformer that receives radar patch encodings at the bottom of the network and cross-attends to optical patch encodings. This approach is similar to CoCa and the original seq2seq transformer design in “Attention Is All You Need”. This explanation is best understood by consulting Figure 1 in our paper.
>
> Please provide additional clarification on what is meant by “freed settings and parameters.” To be clear, all parameters in CROMA are learned end-to-end.
>
> **Q**: *Why invent 2D-ALiBi?*
>
> **A**: ALiBi (ICLR ‘22) is a SoTA relative position encoding method for transformers modeling 1D inputs. Its primary advantage is it can be trained on sequences of, say, 1024 tokens and make inferences on much longer sequences without needing to be re-trained or finetuned on that longer sequence. Because the sizes of imagery in remote sensing datasets vary considerably, we feel that foundation models would benefit from being able to process images with varying numbers of patches. We show how 2D-ALiBi outperforms a SoTA relative position encoding method for ViTs, called PEG (ICLR ‘23), that leverages a convolution between the 1st and 2nd ViT layers and discards position embeddings that are typically added to patch embeddings at the bottom of the network. Furthermore, a model that can flexibly handle images of different sizes can offer a superior trade-off between compute and performance. To illustrate this, we finetune CROMA-B and SatMAE-B on EuroSat for 50 epochs on **96x96px** images—each model is trained once using the default hyper-parameters of each model. Then, we evaluate each model on the EuroSAT validation set at various resolutions:
>
> | Model	|  Test Resolution -> | 32x32 | 64x64 | 96x96 | 120x120 | 224x224 |
> | -------- | ------- | ------- | ------- | ------- | ------- | ------- |
> | SatMAE-B  | 	| 64.1% | 84.2% | 99.1% | 98.3% | 61.4% |
> | CROMA-B |  	| 90.7% | 97.6% | 99.2% | 98.3% | 83.1% |
>
> This demonstrates that CROMA is more robust to differences between train/test-inference image sizes. This robustness permits users to select more compute-efficient image sizes without dramatically sacrificing performance. For instance, if a user requires fast predictions only possible on 32x32px images, CROMA will outperform SatMAE by 26.6% on EuroSAT.
>
> Beyond extrapolating to larger images, 2D-ALiBi outperforms 2D-sinusoidal embeddings and PEG without extrapolation, i.e., testing on the exact resolution as training. We believe this is due to the true relative position encoding of 2D-ALiBi, which is both translation and rotation equivariant. These properties complement data in the remote sensing domain that is overhead imagery whose representations should be translation and rotation invariant.
>
> **Q**: *Why a 75% mask ratio?*
>
> **A**: We select our hyperparameters by considering both performance and cost. As shown in Table 5, a mask ratio of 50% outperforms a mask ratio of 75% by 0.1%, averaged across 6 evaluations. But a 50% mask ratio is 1.8x slower to train due to the encoders processing more patches. This highlights a benefit of CROMA—it is robust to the choice of mask ratio.
>
> **Q**: *Experimental settings.*
>
> **A**: We provide experimental conditions in our appendix, as we focused on areas of most interest to a NeurIPS audience in our main text. We also anonymously share all code, pretrained models, and preprocessed datasets. Please see the appendix.
>
> **Q**: *Model comparisons.*
>
> **A**: We compare CROMA to all relevant foundation models in our application. Furthermore, some of our ablations are equivalent to other SoTA methods that have yet to be explored for remote sensing. For example, only leveraging contrastive learning amounts to adapting Fast Language Image Pretraining [124] (FLIP, published at CVPR ‘23) to our domain, as FLIP performs cross-modal contrastive learning with masked-out samples. An ablation in our appendix with VICReg and a patch-wise invariance loss objective is equivalent to adapting VICRegL [143] (NeurIPS ‘22) to our domain. In our revision, we will clarify the connection between these ablations and other SoTA computer vision methods not yet explored in remote sensing.
>
> **Q**: *Discussion of experimental results.*
>
> **A**: We agree that our submission would benefit from a more detailed discussion of the results. Our main text focused on using our space to validate CROMA as thoroughly as possible experimentally—hence the probing, kNN, K-means, and segmentation experiments. If our paper is accepted, we will have more room to incorporate a deeper discussion in a revised version.

---

> > ### Comment · Reviewer_vsHk · 2023-08-20
> > **Thanks for the reply**
> >
> > Thanks the authors for their reply. Although parts of the issues have been modified, the novelty of this work is limited to NIPS. In addition to the poor organization and written, the contributions of this manuscript is narrowed. Thus, I insist on my original decision, i.e., Reject.

---

### Official Review · Reviewer_too9 · 2023-07-04

**Soundness:** 2 fair
**Presentation:** 2 fair
**Contribution:** 2 fair
**Rating:** 4
**Confidence:** 5

**Summary:**

This paper proposes CROMA to align optical and SAR modal images via contrastive learning and reconstruction. Comprehensive experiments on three datasets have demonstrated the effectiveness of CROMA.

**Strengths:**

1. This paper introduces a multi-modal representation using contrastive learning and reconstruction.
2. The proposed CROMA has exceeded the SatMAE, which only used unimodal images.
3. CROMA is more faster and effective than SatMAE.


**Weaknesses:**

1. CROMA only constructs pos-neg samples from different modal images. Why are image patches in different regions of the same modality not used as negative samples?
2. There miss lots of important details. For example, the number of positive and negative samples is not discussed.
3. How is the sampling ratio of positive and negative samples affected? What is the relationship between positive and negative sample sampling and effective inputs in reconstruction?
4. I am worried about the theoretical innovation of the paper. The paper mainly focuses on the application of contrastive learning loss and reconstruction loss to remote sensing multimodal modeling, which is also very common in medical multimodal and multitemporal. As an extension of SatMAE, I don't know whether the innovation of the paper is enough for NeurIPS. Because there already exists many related papers for remote sensing[1][2].
[1] Ayush K, Uzkent B, Meng C, et al. Geography-aware self-supervised learning[C]//Proceedings of the IEEE/CVF International Conference on Computer Vision. 2021: 10181-10190.
[2] Manas O, Lacoste A, Giró-i-Nieto X, et al. Seasonal contrast: Unsupervised pre-training from uncurated remote sensing data[C]//Proceedings of the IEEE/CVF International Conference on Computer Vision. 2021: 9414-9423.


**Questions:**

N/A

---

> ### Author Rebuttal · Authors · 2023-08-10
>
> Thank you for your thoughtful comments on our work. We appreciate the recognition of the strengths of our work: the introduction of learning multimodal representations by jointly leveraging reconstruction and contrastive learning and the superiority over the current SoTA. We believe we can answer the concerns raised, but we’d like some clarification on one question. Please see our replies below:
>
> **Q**: *Why not use samples from the same modality as negatives in contrastive learning?*
>
> **A**: Please see our response to reviewer iZ73, "**Q**: *Negative optical samples in contrastive learning?*"
>
> **Q**: *Clarification on the number of positives and negatives.*
>
> **A**: We believe that the paragraph starting at L203 discusses this, including the equation below L207—but we acknowledge this could have been more clear. We use the most popular contrastive learning framework in the literature; this uses one positive sample (in our case, matching radar-optical representations), and negatives are all other samples from the batch (in our case, mismatching radar-optical representations)—SimCLR [99] best exemplifies this method. For CROMA-B, we use a batch size of 7200 (the largest that can fit onto a DGX server consisting of 640 GB of VRAM using bfloat16 precision); this means we use 7199 negatives. If our paper is accepted, we will have more space (with a 10th page) to clarify this in our main text.
>
> **Q**: *“How is the sampling ratio of positive and negative samples affected? What is the relationship between positive and negative sample sampling and effective inputs in reconstruction?”*
>
> **A**: We are not sure what is meant by these questions and would appreciate some clarification.
>
> **Q**: *Clarification on the novelty of our CROMA self-supervised learning algorithm.*
>
> **A**: We agree that the main innovation of our paper is combining reconstruction and contrastive losses for learning multimodal representations. Since these two objectives learn different types of representations, we believed combining them would learn more general representations than alone. And we demonstrate this experimentally in our ablations. We have outlined the related work in our paper, but admittedly, it was dense—if our paper is accepted, we will have an extra page of space to further clarify how CROMA differs from related methods. The vast majority of work that leverages reconstruction and contrastive learning to learn joint multimodal representations is in the image-text domain. For example, ALBEF [107] (NeurIPS ‘21), TCL [108] (CVPR ‘22), BLIP [110] (ICML ‘22), MaMMUT [111] (TMLR ‘23) and CoCa [49] (TMLR ‘22)—these were all clever iterations on prior work. In general, these frameworks independently encode full images (no masking) and masked text, then perform a contrastive objective between matching cross-modal samples. Next, they reconstruct the masked text (either with a masked language modeling or autoregressive language modeling objective) conditional on image representations. But none of these approaches were designed for 2D multimodal data. We are primarily inspired by CoCa and significantly adapt it to 2D multimodal data. The simplest way to adapt these frameworks to 2D data would have been to replace the text encoder with another ViT—but encoding mask tokens in a ViT hurts performance and is much slower than hiding patches and leveraging a decoder to predict the hidden patches (this is the main innovation of MAE [31] (CVPR ‘22)). Thus our CROMA framework can best be considered a combination of CoCa and MAE. By leveraging MAE-style masking, we save on compute and provide a target for reconstruction; the efficiency of performing contrastive learning on masked-out cross-modal samples was also found in concurrent work, FLIP [123] (CVPR ‘23). These innovations are significant and justify the broad audience that NeurIPS attracts. Two papers came to similar conclusions in parallel with us, CAV-MAE [50] (ICLR ‘23) and MaViL [52](available on arxiv, also under review), but these papers are in the audio-visual domain and are not designed for spatially aligned data that is ubiquitous to RS and other applications. Our multimodal decoder is designed explicitly for spatially aligned multimodal data as tokens in the decoder predict both hidden radar and optical patches corresponding to a precise location on the ground—this is not possible in the CAV-MAE or MaViL frameworks.
>
> Additionally, with X-ALiBi, we bias the cross-attention matrix based on the relative locations between cross-modal patches—this is only possible with spatially registered data and is the first time position encoding has ever been leveraged in cross-attention. Finally, by providing true relative position encoding that is both translation and rotation equivariant, 2D-ALiBi outperforms the SoTA in position encoding for ViTs, PEG [118] (ICLR ‘23). ALiBi [119] (ICLR ‘22) is a groundbreaking method in NLP that is now the method of choice in many SoTA LLMs because of its extrapolation abilities—an extension of it to 2D data is also novel.
>
> Overall, CROMA builds and improves on methodology recently published in NeurIPS and conferences with similar audiences and impact. Not to mention, CROMA substantially outperforms SatMAE [26] (NeurIPS ‘22) and provides a more thorough evaluation of learned representations. We thank you for asking this question, as it demonstrates that the novelty of CROMA was not sufficiently articulated—we will integrate this discussion in our revision.

---

> ### Comment · Reviewer_too9 · 2023-08-17
> **Thanks for the rebuttal**
>
> Thank the authors for their rebuttal. It partly addressed my concerns, except for the novelty.
> This is a good practice to combine contrastive learning and masked image modeling on multi-modal satellite images; however, I cannot regard it as an innovative approach. There is little knowledge improvement for me.
>
> Authors continually emphasize that their approach goes beyond SatMAE, which is unfair due to different pre-training data and improper baseline, i.e., SSL4EO v.s. fMoW-Sentinel. The pre-training data should be aligned. The right baseline should be SatMAE+ contrastive learning to illustrate your combination is non-trivial; otherwise, it cannot convince most readers in NeurlPS.
>
> I appreciate the extensive empirical results from this manuscript. However, I cannot recommend this manuscript be accepted by NeurlPS at this round, given its limited novelty, unfair comparison, and insufficient theoretical support.

---

> > ### Author Response · Authors · 2023-08-17
> >
> > Thank you for your continued engagement in our work.
> >
> > Regarding novelty, our method is a novel combination of existing methods: cross-modal contrastive learning, multimodal masked autoencoding, and attention with linear biases (ALiBi). Our method was motivated by the intuition that we describe in our paper: (i) that contrastive and reconstructive pretraining objectives learn different representations that might be complementary when combined and (ii) that EO data would benefit from rotation and translation invariant relative position encoding. This research framework—that uses intuition to combine existing methods in novel ways—is strongly represented in NeurIPS every year.
> >
> > Regarding comparisons, we extensively compare CROMA to all foundation models for EO. There are many recent frameworks invented in computer vision that we could leverage to pretrain models on the SSL4EO dataset—but doing so for all new approaches is not practical. Specifically, two concerns are raised: (i) we do not compare to a SatMAE model pretrained on SSL4EO, and (ii) we do not compare to a “SatMAE + contrastive learning” framework. Regarding the first concern, we do not believe that SatMAE pretrained on SSL4EO would improve SatMAE’s performance on benchmarks because, qualitatively, the data distribution of fMoW-Sentinel is closer to these benchmarks than SSL4EO (primarily, the sizes of images). In fact, CROMA outperforms SatMAE when finetuning on the fMoW-Sentinel dataset—this demonstrates that our framework learns better representations than SatMAE. Regarding the second concern, the “SatMAE + contrastive learning” framework does not exist, but we could try it. However, we do not expect it to outperform CROMA because CROMA outperforms VICRegL (see our appendix), which outperforms unimodal MAE + contrastive learning (see the VICRegL paper).
> >
> > We would have been happy to address these two concerns with experiments during the rebuttal period, but these concerns were not raised during the original review. Overall, we are disappointed that these new concerns dropped our score from a borderline accept to a borderline reject.

---

### Official Review · Reviewer_iZ73 · 2023-07-04

**Soundness:** 3 good
**Presentation:** 4 excellent
**Contribution:** 2 fair
**Rating:** 6
**Confidence:** 5

**Summary:**

The paper presents a self supervised representation learning model for multimodal sentinel images. The model learns from geographically aligned optical and radar (sentinel-2 and sentinel-1, respectively) representations that are then used for downstream tasks, such as classification and segmentation. In the paper, classification and segmentation tasks are illustrated by using different approaches, namely, fine tuning, linear probing and nonlinear probing (MLP). Authors also show other quantitative evaluations (knn, kmeans over classes, a UMAP) to show the quality of the learned representation over SatMAE [26], which is the main competing method.


**Strengths:**

- The paper deals with an important topic, and the models and results presented in the paper are significant for a variety of applications making use of sentinel data.
- Results are validated on well known datasets in the field, and show superiority over a series of strong baselines and several metrics. Ablation study is very complete and shows how the model performs under different changes in the modules.
- The approach of combining masked out reconstruction and contrastive losses in learning SSL representations is, to the best of my knowledge, novel in the field of geographically aligned data. The use of different modalities is also interesting, although the synergistic use of optical and radar images is well known and studied, but for specific applications.
- The paper is very dense but clear enough, well written and well structured.


**Weaknesses:**

I think that the paper is sound and although it touches upon a niche application of computer vision that might not be of wide interest for the NeurIPS audience, it could be a good contribution. However I have a series of comments that I think could be addressed and improve the paper. In general:
- The data description and the explanation of the different levels of preprocessing for each dataset (eg atmospheric corrections from L1C to L2A), and how those influence the model, are not well explained. For instance, sentinel 2 has 13 channels, but of which only a subset are useful for land cover applications. Also, spatial resolution of the different S2 channels is very different (from 10 to 60m) and the one from S1 changes depending on the processing levels.
- The different benchmark tested are characterised by different preprocessing and channel subsets, and it is unclear how the models are fine tuned in this setting and what is the dependency on those aspects.
- I felt that the related work section at L72 makes plenty of references but it is not so good at clarifying some main lines of research, pros and cons of those. There are many papers, maybe too many, and it is hard to get some information out of that.

**Questions:**


- L91: I found the concept of "optionally multimodal" very interesting, and I think it could be better framed. I am not sure whether the proposed CROMA it is indeed optionally multimodal, but the fact that some image pairs are not available concurrently is a very common issues which is often solved by temporal composition, but might not be optimal for some fine grained monitoring applications.
- L109: it is not completely clear to me why (beyond the ablations) RPE offer, sometimes, good performance when extrapolating over larger images. At the same time, i think, contrarily what stated afterwards, the wild variability of remote sensing images is not an issue: images are referenced absolutely to geographical coordinates and can be cropped following the constraints of the task and hardware.
- What is not so clear to me, is how the choice of spectral channels for S2 is done, and how resolution is dealt with. S2 channels have a ground sampling distance that varies between 10m, 20m and 60m, while radar can vary depending on the processing level. It is unclear how patches are sampled and how the mismatches in resolution is dealt with, since if sampling eg a 80x80 pixels patch, the actual content varies a lot unless interpolated and upsampled, which could introduce artefacts in itself. I think these data preprocessing steps should be better explained.
- It is unclear at which level of processing the data is used, whether L2A corrected or L1C Sentinel-2 data, and if so, how the processing is performed. It is also mentioned that 12 bands are used, but in facts S2 has 13. Just that some of these bands are used for sensing properties of the atmosphere such as clouds and aerosols, and do not help in performing land-cover / land-use related modelling. Again, I think that some of these aspects should be clarified in the main text.
- I wonder why [86] has not been included in the baselines, as it is one of the main papers highlighted in the related work section dedicated to RS representaiotns.
- In light of the above points, some datasets used to test the model have very different properties and characteristics. how is the CROMA model pretrained on 12 S2 channels, retrained for each of the benchmark making use of the specific data provided? eg the fMoW, as far as i remember, only make use of 8 channel and not 12.
- L203 and following: It is unclear why no other optical image participates in the definition of negative samples, this could be maybe beneficial to encode differences in landcover at different locations, or account for seasonality effects.
- L240: it is unclear what "single label benchmarks" are.





**Limitations:**

- Limitations shortly mentioned in the conclusion section. I agree with those highlighted.
- I had the feeling when reading the paper, the fact that CROMA was performing better than SatMAE, that is all it was needed. I think that SatMAE comes with pros and cons that are not very well discussed and framed. I think that since the paper is comparing and improving directly upon SatMAE, these aspects could have been better presented.
- I really missed a test on non Sentinel data. I do agree that Sentinel is a great data source, but it is not the only source used, particular when studies need to go back before 2015/2016. It would have been nice to see some results on other satellite data, but I understand that this could have been too much work or out of scope, but I would nonetheless mention it (this goes beyond higher spatial and spectral resolution, it is often interesting to transfer models to lower spatial and spectral resolution).

---

> ### Author Rebuttal · Authors · 2023-08-09
>
> Thank you for your thoughtful comments on our work. We believe we can address all the concerns you raise.
>
> **Q**: *Optionally multimodal.*
>
> **A**: CROMA is indeed optionally multimodal. In sections 4.1 & 4.2, we only use the optical encoder (Sentinel-2-only tasks). In section 4.3, we use all three encoders (multimodal tasks). And in section 5.1, we use all three combinations (optical-only, radar-only, and joint). Crucially, each of the three combinations of encoders requires a single pretraining run that jointly pretrains these three encoders end-to-end.
>
> **Q**: *RPE.*
>
> **A**: RPE captures the relative locations between tokens in a transformer rather than their absolute locations in the sequence. As ALiBi [119] (ICLR ‘22) shows, RPE methods often fail to extrapolate to longer sequences effectively. PEG (ICLR ‘23) showed that a simple convolution between the 1st and 2nd ViT layers (along with discarding position embeddings) improves performance and enables extrapolation. For our application, we show that 2D-ALiBi outperforms PEG with and without extrapolation. 2D-ALiBi is a true RPE method that is both translation and rotation equivariant, these properties are desirable for EO imagery.
>
> The sizes of imagery in RS benchmarks vary considerably. Still, we agree that users can query and crop imagery to their liking if they have access to the original data sources. Besides this, image-size extrapolation would be a significant feature for RS foundation models. By increasing or decreasing the number of patches, users can vary the amount of compute they wish to spend on a sample. To illustrate this, we finetune CROMA-B and SatMAE-B on EuroSat for 50 epochs on **96x96px** images—each model is trained once using the default hyper-parameters of each model. Then, we evaluate each model on the EuroSAT validation set at various resolutions:
>
> | Model	|  Test Resolution -> | 32x32 | 64x64 | 96x96 | 120x120 | 224x224 |
> | -------- | ------- | ------- | ------- | ------- | ------- | ------- |
> | SatMAE-B  | 	| 64.1% | 84.2% | 99.1% | 98.3% | 61.4% |
> | CROMA-B |  	| 90.7% | 97.6% | 99.2% | 98.3% | 83.1% |
>
> For instance, if a user requires fast predictions only possible on 32x32px images, CROMA will outperform SatMAE by 26.6% on EuroSAT.
>
> **Q**: Data preprocessing.
>
> **A**: For pretraining, we use the SSL4EO dataset assembled, preprocessed, and published by another lab [85]. This dataset provides paired Sentinel-2 L2A data (atmospheric corrections removing the cirrus band, resulting in 12 bands) and Sentinel-1 GRD data—this is briefly stated on L220 of our paper. The data is already upsampled to a 10m per pixel spatial resolution for all relevant channels. According to the SSL4EO codebase, using Google Earth Engine, they query: (i) the 'COPERNICUS/S2_SR' collection, filtering images with greater than 10% cloud coverage, and bilinear spatial upsampling, and (ii) the 'COPERNICUS/S1_GRD' collection in interferometric wide mode with VV and VH channels. We selected the SSL4EO dataset for pretraining because it is the largest preprocessed Sentinel-1 & 2 imagery collection and can be easily downloaded, making replication feasible. Furthermore, given the data collected by SSL4EO, before feeding them into our models, we normalize all channels exactly like SatMAE [26]. In our revised appendix we will clarify how data is preprocessed.
>
> **Q**: *Handling 13 channels.*
>
> **A**: The Sentinel-2 benchmarks contain either 12 (with cirrus removed) or 13 channels (with cirrus included). When applying CROMA to benchmarks that contain 13 channels, we drop the cirrus band, giving our model 12 of the 13 available channels. We do not re-train CROMA for each benchmark—all CROMA models (CROMA-B, CROMA-L, and ablations) are pretrained on the SSL4EO dataset described above. fMoW-Sentinel contains 13 channels [26].
>
> **Q**: *SatViT-V2 comparison.*
>
> **A**: SatViT-V2 [86] only processes Sentinel-1 & 2 data stacked along the channel dimension—it is not optionally multimodal. Therefore, it cannot process the optical-only datasets we evaluate in sections 4.1 & 4.2. In section 4.3, we compare CROMA with SatViT-V2 on two multimodal benchmarks, BigEarthNet [126] and DFC2020 [137]—CROMA significantly outperforms SatViT-V2, which was pretrained using masked autoencoding.
>
> **Q**: *Negative optical samples in contrastive learning?*
>
> **A**: We selected this cross-modal contrastive learning (CMCL) framework because it is the standard in the literature. The representations of non-matching samples of the same modality are much closer to each other than the representations of matching samples of different modalities. This has been widely observed [*1, *2, *3] and is now coined the “modality gap”—nicely illustrated in Figure 1 of “Mind the Gap” [*1]. We observe this same modality gap between optical and radar representations in CROMA (both at initialization and convergence). Including optical samples as negatives in the CMCL calculation amounts to introducing very difficult negatives. Although “hard negatives” can improve representations, overly difficult negatives can hurt representations [*4, 36, 145].
>
> In our appendix, we did experiment with hard negatives via hard negative mixing (HNM, [145])—this mixes optical representations with radar representations to build hard negatives. We show HNM hurts representations across all 6 evaluations. Using optical representations as negatives would create even more difficult negatives than mixed optical and radar representations.
>
> [*1] “Mind the Gap: Understanding the Modality Gap in Multi-modal Contrastive Representation Learning,” in NeurIPS ‘22
>
> [*2] “Towards understanding the modality gap in CLIP,” in ICLR ‘23 Workshop
>
> [*3] “Understanding and constructing latent modality structures in multi-modal representation learning,” in CVPR ‘23
>
> [*4] “Contrastive learning with hard negative samples,” in ICLR ‘21

---

> > ### Comment · Reviewer_iZ73 · 2023-08-14
> > **Follow up**
> >
> > I'd like to thank the Authors for the follow up.
> >
> > From my perspective, the rebuttal is clear, and definitely improves my understanding of the paper. There are still a couple of minor points that are not explicitly addressed, but I agree those are not worthy discussion at this stage and can be directly incorporated in the camera ready.
> >
> > I think the contribution is interesting, and overall relevant not only to the EO community, so I am happy to increase the score to weak accept, and looking forward to discuss further with other reviewers, if needed.

---

### Author Rebuttal · Authors · 2023-08-10

We thank all reviewers for their feedback. Before replying to reviewers individually we will restate our contributions:

* We leverage reconstructive and contrastive objectives to learn joint multimodal representations. This is not only novel for Earth Observation (EO) but is novel for multimodal 2D data. Of the many multimodal learning frameworks recently published, only two also learn multimodal representations of 2D data via these combined pretraining objectives: CAV-MAE [50] (ICLR ‘23) and MaViL [52] (available on arxiv, also under review). Both CAV-MAE and MaViL were developed concurrently with our work. Both frameworks are designed for audio-visual signals, not spatially aligned sensor data that we focus on. This feature lets us fuse corresponding cross-modal patches spatially—learning joint multimodal patch representations. We are glad that reviewers iZ73, too9, and FExC acknowledge this contribution.
* We extend ALiBi [119] (ICLR ‘22) to 2D data and show that 2D-ALiBi outperforms PEG [118] (ICLR ‘23) for our application and is novel to ViTs. Additionally, our X-ALiBi method is the first time position encoding has been leveraged in cross-attention, not just in our application. These methods allow CROMA to effectively generalize to smaller or larger images at test-time without further training. This means that CROMA models—and ViTs that will leverage our work by using 2D-ALiBi—offer a superior trade-off between accuracy and compute. We are glad that reviewer FExC “really likes” this contribution and seems interested in building on it.
* We thoroughly demonstrate that CROMA significantly outperforms the previous SoTA, SatMAE [26] (NeurIPS ‘22). Our evaluation consists of finetuning, linear probing, nonlinear probing, kNN classifying, and K-means clustering the learned representations of CROMA and all other SoTA foundation models for Earth Observation under identical conditions. Additionally, two of our ablations are equivalent to adapting other SoTA algorithms in computer vision to our domain: (i) Fast Language Image Pretraining (FLIP [124], published at CVPR ‘23) performs cross-modal contrastive learning with masked-out samples to efficiently outperform CLIP [124], and (ii) VICRegL [143] (NeurIPS ‘22) combines the VICReg [142] objective between image representations and an MSE patch-wise invariance objective to learn SoTA local representations (see this experiment in our appendix). CROMA outperforms both. We are glad that reviewers iZ73, too9, and FExC view our extensive experiments and ablations as a strength.

We acknowledge that our paper is dense and are glad reviewers iZ73 and FExC liked our writing and presentation. Our extensive experiments and ablations were only made possible by this density and by assuming our readers had a prior understanding of related work—specifically, vision transformers, masked autoencoders, contrastive learning, and prior foundation models for EO. Without this background, we completely understand how our paper may be challenging to comprehend. Should our paper be accepted to this conference, we will be granted a 10th page which we will dedicate to providing more background, clarifying our contributions, and including more experimental conditions in our main text (as of now, much of our experimental conditions are in the appendix).

Our paper improves on methods recently published at this conference and conferences with similar audiences and impact. As reviewer FExC correctly points out, many other domains feature multiple sensors whose data are spatially registered. Our framework can be directly used for these applications. We also see tremendous potential for X- and 2D-ALiBi, (or variants building on them, e.g. 3D data) that can be broadly applied to transformers. We are targeting this conference to publish our work for these reasons, not only because we outperform SatMAE.

We are happy to continue our discussions with all reviewers.

---

### Decision · Program_Chairs · 2023-09-21

**Decision:**

Accept (poster)

**Comment:**

This paper introduces a framework for self-supervised, multi-modal remote sensing, showing that it outperforms SatMAE on a range of tasks. While there is disagreement among the reviewers, the main objection raised is "novelty" which is difficult to evaluate in settings where application-specific challenges lead to innovations that adapt existing ML tools. Of the two dissenting reviewers, one lowered their review from a borderline accept after raising a request for new experiments only at the end of the discussion phase, when authors were unable to respond. The other dissenting review did not provide sufficient detail or support for their concerns. Accordingly, I recommend acceptance.